# Nearly Optimal Approximation of Matrix Functions by the Lanczos Method

Noah Amsel[*]  Tyler Chen[*]  Anne Greenbaum[†]  Cameron Musco[‡]  Christopher Musco[*]

## Abstract

Approximating the action of a matrix function $f(\mathbf{A})$ on a vector $\mathbf{b}$ is an increasingly important primitive in machine learning, data science, and statistics, with applications such as sampling high dimensional Gaussians, Gaussian process regression and Bayesian inference, principle component analysis, and approximating Hessian spectral densities. Over the past decade, a number of algorithms enjoying strong theoretical guarantees have been proposed for this task. Many of the most successful belong to a family of algorithms called *Krylov subspace methods*. Remarkably, a classic Krylov subspace method, called the Lanczos method for matrix functions (Lanczos-FA), frequently outperforms newer methods in practice. Our main result is a theoretical justification for this finding: we show that, for a natural class of *rational functions*, Lanczos-FA matches the error of the best possible Krylov subspace method up to a multiplicative approximation factor. The approximation factor depends on the degree of $f(x)$'s denominator and the condition number of $\mathbf{A}$, but not on the number of iterations $k$. Our result provides a strong justification for the excellent performance of Lanczos-FA, especially on functions that are well approximated by rationals, such as the matrix square root.

## 1 Introduction

Given a symmetric matrix $\mathbf{A} \in \mathbb{R}^{d \times d}$ with eigendecomposition $\mathbf{A} = \sum_{i=1}^{d} \lambda_i \mathbf{u}_i \mathbf{u}_i^\mathsf{T}$, the matrix function $f(\mathbf{A})$ corresponding to a scalar function $f : \mathbb{R} \to \mathbb{R}$ is defined as

$$f(\mathbf{A}) := \sum_{i=1}^{d} f(\lambda_i) \mathbf{u}_i \mathbf{u}_i^\mathsf{T}. \tag{1}$$

Matrix functions arise throughout machine learning, data science, and statistics. For instance, the matrix square root is used in sampling Gaussians, Bayesian modeling, and Gaussian processes [4, 5, 59], general fractional matrix powers are used in Markov chain modeling and Rényi entropy estimation [68, 42, 43, 19], the matrix logarithm is used for determinantal point processes, kernel learning, and approximating log-determinants for Gaussian process regression and Bayesian inference [17, 39, 31], the matrix sign is used in principal components regression and spectral density estimation [29, 18, 44, 58, 32, 69, 13, 7, 14], and the matrix exponential is used in network science [3, 67, 48]. In many of these applications, we do not need to compute the matrix $f(\mathbf{A})$ itself, but rather its action on a vector $\mathbf{b} \in \mathbb{R}^d$; i.e., $f(\mathbf{A})\mathbf{b}$. This task can be performed much more efficiently than computing an eigendecomposition of $\mathbf{A}$ and forming $f(\mathbf{A})$ using (1).

Perhaps the first general purpose method for approximating $f(\mathbf{A})\mathbf{b}$ is the Lanczos method for matrix function approximation (Lanczos-FA) [22, 30], which is the focus of the present paper. Over the past decade, a number of special purpose algorithms, designed for a single function or class of functions,

---

[*]New York University (`noah.amsel@nyu.edu`, `tyler.chen@nyu.edu`, `cmusco@nyu.edu`)

[†]University of Washington (`greenbau@uw.edu`)

[‡]University of Massachusetts Amherst (`cmusco@cs.umass.edu`)

38th Conference on Neural Information Processing Systems (NeurIPS 2024).

have been developed [23, 16, 26, 44, 11]. These newer algorithms often satisfy strong theoretical guarantees better than the best-known bounds for Lanczos-FA. The present paper is motivated by the following remarkable observation:

> Despite being arguably the simplest and most general algorithm for computing $f(\mathbf{A})\mathbf{b}$, *Lanczos-FA frequently outperforms special purpose algorithms*, sometimes by orders of magnitude, on common test problems.

For instance, in Figures 4, 5, and 8 we compare Lanczos-FA with specialized algorithms [44, 59, 11] that satisfy the best-known theoretical guarantees for computing $f(\mathbf{A})\mathbf{b}$ for the particular functions they were designed for. In these experiments Lanczos-FA drastically outperforms these methods, despite the fact that it was not designed for any particular function. Because of its outstanding performance, Lanczos-FA is perhaps the most commonly used algorithm for computing $f(\mathbf{A})\mathbf{b}$ in practice. The main goal of this paper is to improve our theoretical understanding of *why* Lanczos-FA performs so well, in order to help close the theory-practice gap.

## 1.1 Krylov subspace methods

Lanczos-FA falls into a class of algorithms called Krylov Subspace Methods (KSMs). KSMs are among the most powerful and widely used algorithms for a broad range of computational tasks including solving linear systems, computing eigenvalues/vectors, and low-rank approximation [62, 38, 47, 65]. Like other KSMs for computing $f(\mathbf{A})\mathbf{b}$, Lanczos-FA iteratively constructs an approximation from the Krylov subspace

$$\mathcal{K}_k(\mathbf{A}, \mathbf{b}) := \operatorname{span}\{\mathbf{b}, \mathbf{A}\mathbf{b}, \dots, \mathbf{A}^{k-1}\mathbf{b}\}. \tag{2}$$

The Lanczos algorithm [46] produces an orthonormal basis $\mathbf{Q} = [\mathbf{q}_1, \dots, \mathbf{q}_k]$ for the Krylov subspace $\mathcal{K}_k(\mathbf{A}, \mathbf{b})$ and a symmetric tridiagonal matrix $\mathbf{T}$ satisfying $\mathbf{T} = \mathbf{Q}^\mathsf{T}\mathbf{A}\mathbf{Q}$ [62]. The Lanczos-FA algorithm uses this $\mathbf{Q}$ and $\mathbf{T}$ to approximate $f(\mathbf{A})\mathbf{b}$.

**Definition 1.** *The Lanczos-FA iterate for a problem instance* $(f, \mathbf{A}, \mathbf{b}, k)$ *is defined as*

$$\operatorname{lan}_k(f; \mathbf{A}, \mathbf{b}) := \mathbf{Q}f(\mathbf{T})\mathbf{Q}^\mathsf{T}\mathbf{b},$$

*where* $\mathbf{Q}$ *and* $\mathbf{T}$ *are as above.*

In our analysis we assume exact arithmetic. We do not discuss the implementation of Lanczos or Lanczos-FA since there are many resources on this topic; see for instance [52, 8, 9]. Fortunately, if the Lanczos algorithm is implemented properly, its finite-precision behavior closely follows its exact-arithmetic behavior on a nearby problem [33]. See Section 5 for further discussion.

## 1.2 Optimality guarantees for Krylov subspace methods

For a wide variety of problem instances, Lanczos-FA is observed to converge almost as quickly as the *best* approximation to $f(\mathbf{A})\mathbf{b}$ that could be returned by *any* KSM run for the same number of iterations. In particular, all KSMs output approximations that lie in the span of $\mathcal{K}_k(\mathbf{A}, \mathbf{b})$, that is, approximations of the form $p(\mathbf{A})\mathbf{b}$ for a polynomial $p$ of degree less than $k$. Given a problem instance $(f, \mathbf{A}, \mathbf{b}, k)$, the best possible approximation returned by a Krylov method is[4]

$$\operatorname{opt}_k(f; \mathbf{A}, \mathbf{b}) := \operatorname*{argmin}_{\mathbf{x} \in \mathcal{K}_k(\mathbf{A}, \mathbf{b})} \|f(\mathbf{A})\mathbf{b} - \mathbf{x}\|_2.$$

By definition, the error of this Krylov optimal iterate can be characterized as follows:

$$\|f(\mathbf{A})\mathbf{b} - \operatorname{opt}_k(f; \mathbf{A}, \mathbf{b})\|_2 = \min_{\deg(p)<k} \|f(\mathbf{A})\mathbf{b} - p(\mathbf{A})\mathbf{b}\|_2. \tag{3}$$

For general matrix functions, no efficient algorithm for computing $\operatorname{opt}_k(f; \mathbf{A}, \mathbf{b})$ is known, but as shown in Figure 1, the solution returned by Lanczos-FA often nearly matches the error of this best approximation. It is thus natural to ask if, at least for some class of problems, Lanczos-FA satisfies the following strong notion of approximate optimality:

---

[4]We choose to measure error in the Euclidean norm, and thus define optimality with respect to this norm. While other norms have been considered in prior work on matrix-function approximation [11], the Euclidean norm a simple starting point.

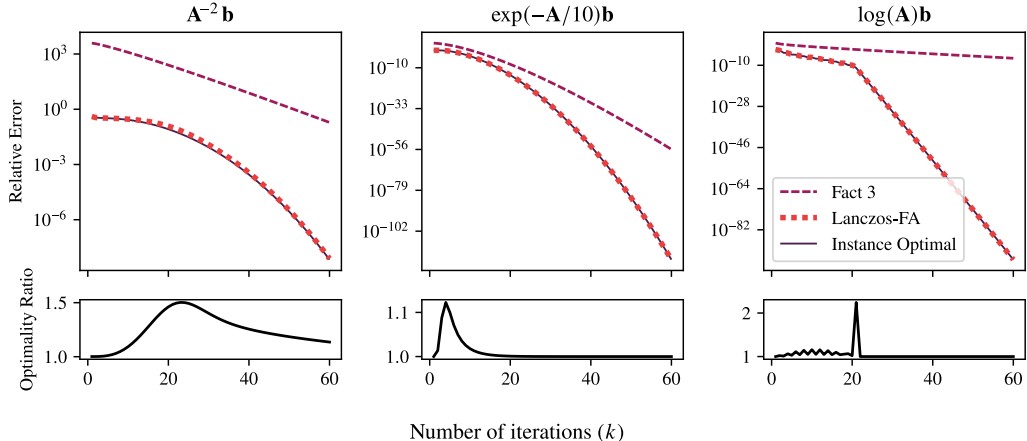

**Figure 1:** Lanczos-FA error $\|f(\mathbf{A}) - \mathsf{lan}_k(f; \mathbf{A}, \mathbf{b})\|_2$ at each iteration for several functions/spectra. "Instance Optimal" is the right hand side of Definition 2 with $C = c = 1$, which is a lower bound for all KSMs, including Lanczos-FA. Lanczos-FA performs nearly instance optimally on a wide range of problems, far better than Fact 3 predicts. This is easily seen in the bottom plots, which show the ratio of the error of the Lanczos-FA iterate and the Krylov optimal iterate, $\mathsf{opt}_k(f; \mathbf{A}, \mathbf{b})$.

**Definition 2** (Near Instance Optimality). *For a problem instance* $(f, \mathbf{A}, \mathbf{b}, k)$, *we say that a Krylov method is nearly instance optimal with parameters $C$ and $c$ if*

$$\|f(\mathbf{A})\mathbf{b} - \mathsf{alg}_k(f; \mathbf{A}, \mathbf{b})\|_2 \leq C \min_{\deg(p) < ck} \|f(\mathbf{A})\mathbf{b} - p(\mathbf{A})\mathbf{b}\|_2.$$

*Above,* $\mathsf{alg}_k(f; \mathbf{A}, \mathbf{b})$ *denotes the output of an algorithm (e.g., Lanczos-FA) obtained from the Krylov subspace* $\mathcal{K}_k(\mathbf{A}, \mathbf{b})$, *i.e., the output after $k$ iterations.*

In Definition 2, $C \geq 1$ and $c \leq 1$ allow some slack in comparing to the Krylov optimal iterate. The right hand side depends on the entire problem instance, $(f; \mathbf{A}, \mathbf{b})$, which is why we call the guarantee "instance optimal".

## 1.3 Existing near-optimality analyses of Lanczos-FA

The best bound for Lanczos-FA applying to a broad class of functions is the following common bound for Lanczos-FA; see for instance [61, 54, 52, 10].

**Fact 3.** *For all problem instances* $(f, \mathbf{A}, \mathbf{b}, k)$, *Lanczos-FA satisfies*

$$\|f(\mathbf{A})\mathbf{b} - \mathsf{alg}_k(f; \mathbf{A}, \mathbf{b})\|_2 \leq 2\|\mathbf{b}\|_2 \min_{\deg(p) < k} \left( \max_{x \in [\lambda_{\min}, \lambda_{\max}]} |f(x) - p(x)| \right).$$

Fact 3 is in some sense an optimality guarantee; it compares the convergence of Lanczos-FA to the best possible *uniform* polynomial approximation to $f(x)$. However, it does not take into account properties of $\mathbf{A}$ such as isolated or clustered eigenvalues, and as seen in Figure 1, it typically only gives a loose upper bound on the performance of Lanczos-FA.

To date, near-instance-optimality guarantees for Lanczos-FA akin to those of Definition 2 are known only in a few special cases. The most well-known is when $f(x) = 1/x$ and $\mathbf{A}$ is positive definite, in which case Lanczos-FA is mathematically equivalent to the celebrated Conjugate Gradient algorithm, and therefore exactly optimal in the $\mathbf{A}$-norm [47]. The instance-optimality guarantee for Lanczos-FA in this setting is used to prove well-known super-exponential convergence in certain settings [6, 47, 8]. In contrast, a bound like Fact 3 only provides exponential convergence and is widely understood by the numerical linear algebra community to not accurately describe the actual behavior of the algorithm in most cases [35, 47, 8]. The only other near-optimality guarantees for Lanczos-FA of which we are aware concern $\mathbf{A}^{-1}\mathbf{b}$ for nonsymmetric $\mathbf{A}$ [12] and the matrix exponential $\exp(-t\mathbf{A})\mathbf{b}$ [20]. In both cases, Lanczos-FA satisfies guarantees that are reminiscent of (although weaker than) Definition 2. Further discussion is given in Appendix B.1.

We also remark that there are a number of works which aim to relate the convergence of of Lanczos-FA for functions with certain integral representations to the convergence of conjugate gradient on linear systems [41, 24, 27, 28, 25, 10]. While these analyses provides spectrum dependent convergence guarantees, they are weaker than Definition 2 because it is not clear how the best possible KSM approximation to a given function relates to the convergence of conjugate gradient. These bounds were mostly developed for use as a posteriori stopping criteria rather than as a theoretical explication for the behavior of Lanczos-FA.

## 1.4 Our contributions

In Section 2, we prove near instance optimality for a broad class of rational functions (Theorem 4). To the best of our knowledge, this is the first true instance-optimality guarantee for Lanczos-FA to be proven for any function besides $f(x) = 1/x$. In Section 2.2, we discuss how results of this kind imply related guarantees for functions that are uniformly well-approximated by rationals, which includes many functions of interest in machine learning. In Section 3, we additionally show that Lanczos-FA satisfies a weaker version of near optimality for two crucial non-rational matrix functions—the square root and inverse square root (Theorems 6 and 7). Appendix C compares the this version of optimality to Definition 2 (near instance optimality) and to that of Fact 3. In Section 4, we present experimental evidence showing that, for many natural problem instances, our bounds are significantly sharper than the standard bound of Fact 3. These experiments also demonstrate that despite its generality, Lanczos-FA often converges significantly faster than other methods in practice. We conclude with a discussion of next steps and open problems in Section 5.

## 2 Near optimality for rational functions

In this section, we study the Lanczos-FA approximation to $r(\mathbf{A})\mathbf{b}$, where $r(x)$ is a rational function with real-valued poles that lie in $\mathbb{R} \setminus \mathcal{I}$, where $\mathcal{I} := [\lambda_{\min}, \lambda_{\max}]$. Specifically,

$$r(x) := n(x)/m(x), \tag{4}$$

where $n(x)$ is any degree $p$ polynomial and

$$m(x) = (x - z_1)(x - z_2) \cdots (x - z_q), \qquad z_i \in \mathbb{R} \setminus \mathcal{I}.$$

Since $z_j \notin \mathcal{I}$, $\pm(\mathbf{A} - z_j\mathbf{I})$ is either positive definite or negative definite. For convenience, we define

$$\mathbf{A}_j := \begin{cases} +(\mathbf{A} - z_j\mathbf{I}) & z_j < \lambda_{\min} \\ -(\mathbf{A} - z_j\mathbf{I}) & z_j > \lambda_{\max} \end{cases}. \tag{5}$$

Our main result on rational functions is the following near-instance-optimality bound, which holds under a mild assumption on the number of iterations $k$:

**Theorem 4.** *Let* $r(x) = n(x)/m(x)$ *be a degree* $(p, q)$-*rational function as in (4) and define* $\mathbf{A}_j$ *as in (5). Then, if* $k > \max\{p, q - 1\}$, *the Lanczos-FA iterate satisfies the bound*

$$\|r(\mathbf{A})\mathbf{b} - \mathsf{lan}_k(r; \mathbf{A}, \mathbf{b})\|_2 \leq q \cdot \kappa(\mathbf{A}_1) \cdot \kappa(\mathbf{A}_2) \cdots \kappa(\mathbf{A}_q) \min_{\deg(p) < k - q + 1} \|r(\mathbf{A})\mathbf{b} - p(\mathbf{A})\mathbf{b}\|_2.$$

We prove this theorem in Appendix A. Above, $\kappa(\mathbf{A}_i)$ is the condition number of $\mathbf{A}_i$, the ratio of the largest to smallest magnitude eigenvalues of $\mathbf{A}_i$. Theorem 4 shows that Lanczos-FA used to approximate $r(\mathbf{A})\mathbf{b}$ satisfies near instance optimality, as in Definition 2, with

$$C = q \cdot \kappa(\mathbf{A}_1) \cdot \kappa(\mathbf{A}_2) \cdots \kappa(\mathbf{A}_q), \qquad c = 1 - (q - 1)/k.$$

In particular, when $\mathbf{A}$ is positive definite and each of the $z_i$ are negative (as is the case for rational function approximations of many functions including the square root [37]), then $C \leq q\kappa(\mathbf{A})^q$.

As discussed in Section 5, we expect that Theorem 4 can be tightened by significantly reducing the prefactor. Nevertheless, as illustrated in Figure 2, even in its current form, the bound improves on the standard bound of Fact 3 in many natural cases; i.e., it more tightly characterizes the observed convergence of Lanczos-FA. Finally, as discussed further in Section 2.2, we note that, beyond rational functions being an interesting function class in their own right, Theorem 4 has implications for understanding the convergence of Lanczos-FA for functions like the square root which are much easier to approximate with rational functions than polynomials.

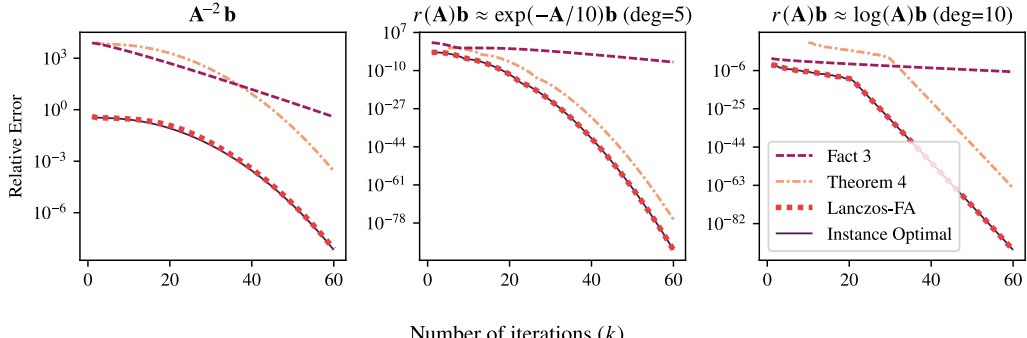

**Figure 2:** Despite its large prefactor, the bound of Theorem 4 qualitatively captures the convergence behavior of Lanczos-FA for rational functions. It can be tighter than the standard bound of Fact 3, even for a moderate number of iterations $k$. We use rational approximations to $\exp(-x/10)$ and $\log(x)$ for comparison with Figure 1; see Section 4 for more details.

### 2.1 Proof Sketch

Our proof of Theorem 4 leverages the near optimality of Lanczos-FA in computing $\mathbf{A}^{-1}\mathbf{b}$ to obtain a bound for general rational functions. For illustration, consider the simplest possible rational function for which we are unaware of any previous near-optimality bounds: $\mathbf{A}^{-2}\mathbf{b}$ when $\mathbf{A}$ is positive definite. The Lanczos-FA approximation for this function is $\mathbf{Q}\mathbf{T}^{-2}\mathbf{Q}^{\mathsf{T}}\mathbf{b} = \mathbf{Q}\mathbf{T}^{-1}\mathbf{Q}^{\mathsf{T}}\mathbf{Q}\mathbf{T}^{-1}\mathbf{Q}^{\mathsf{T}}\mathbf{b}$. Using the triangle inequality and submultiplicativity, we can bound the error of the approximation as

$$\|\mathbf{A}^{-2}\mathbf{b} - \mathsf{lan}_k(x^{-2}; \mathbf{A}, \mathbf{b})\|_2$$
$$\leq \|\mathbf{A}^{-2}\mathbf{b} - \mathbf{Q}\mathbf{T}^{-1}\mathbf{Q}^{\mathsf{T}}\mathbf{A}^{-1}\mathbf{b}\|_2 + \|\mathbf{Q}\mathbf{T}^{-1}\mathbf{Q}^{\mathsf{T}}\mathbf{A}^{-1}\mathbf{b} - \mathbf{Q}\mathbf{T}^{-2}\mathbf{Q}^{\mathsf{T}}\mathbf{b}\|_2$$
$$\leq \|\mathbf{A}^{-2}\mathbf{b} - \mathbf{Q}\mathbf{T}^{-1}\mathbf{Q}^{\mathsf{T}}\mathbf{A}\mathbf{A}^{-2}\mathbf{b}\|_2 + \|\mathbf{Q}\mathbf{T}^{-1}\mathbf{Q}^{\mathsf{T}}\|_2 \cdot \|\mathbf{A}^{-1}\mathbf{b} - \mathbf{Q}\mathbf{T}^{-1}\mathbf{Q}^{\mathsf{T}}\mathbf{A}\mathbf{A}^{-1}\mathbf{b}\|_2. \quad (6)$$

Using the normal equations and the fact that $\mathbf{Q}$ is a basis of the Krylov subspace, it is possible to show that $\mathbf{Q}\mathbf{T}^{-1}\mathbf{Q}^{\mathsf{T}}\mathbf{A} = \mathbf{Q}(\mathbf{Q}^{\mathsf{T}}\mathbf{A}\mathbf{Q})^{-1}\mathbf{Q}^{\mathsf{T}}\mathbf{A}$ is the $\mathbf{A}$-norm[5] projector onto Krylov subspace. Hence

$$\|\mathbf{A}^{-2}\mathbf{b} - \mathbf{Q}\mathbf{T}^{-1}\mathbf{Q}^{\mathsf{T}}\mathbf{A}\mathbf{A}^{-2}\mathbf{b}\|_{\mathbf{A}} = \min_{\mathbf{x} \in \mathcal{K}_k(\mathbf{A}, \mathbf{b})} \|\mathbf{A}^{-2}\mathbf{b} - \mathbf{x}\|_{\mathbf{A}}$$
$$:= \min_{\deg(p) < k} \|\mathbf{A}^{-2}\mathbf{b} - p(\mathbf{A})\mathbf{b}\|_{\mathbf{A}}.$$

Therefore, using $\|\mathbf{x}\|_2 \leq (1/\sqrt{\lambda_{\min}})\|\mathbf{x}\|_{\mathbf{A}}$ and $\|\mathbf{x}\|_{\mathbf{A}} \leq \sqrt{\lambda_{\max}}\|\mathbf{x}\|_2$, the first term on the right hand side of (6) can be bounded as

$$\|\mathbf{A}^{-2}\mathbf{b} - \mathbf{Q}\mathbf{T}^{-1}\mathbf{Q}^{\mathsf{T}}\mathbf{A}\mathbf{A}^{-2}\mathbf{b}\|_2 \leq \sqrt{\kappa(\mathbf{A})} \min_{\deg(p) < k} \|\mathbf{A}^{-2}\mathbf{b} - p(\mathbf{A})\mathbf{b}\|_2.$$

Similarly, the second factor of the second term on the right hand side of (6) can be bounded as

$$\|\mathbf{A}^{-1}\mathbf{b} - \mathbf{Q}\mathbf{T}^{-1}\mathbf{Q}^{\mathsf{T}}\mathbf{A}\mathbf{A}^{-1}\mathbf{b}\|_2 \leq \sqrt{\kappa(\mathbf{A})} \min_{\deg(p) < k} \|\mathbf{A}^{-1}\mathbf{b} - p(\mathbf{A})\mathbf{b}\|_2.$$

Then, since $xp(x)$ is polynomial of one degree larger than $p(x)$,

$$\min_{\deg(p) < k} \|\mathbf{A}^{-1}\mathbf{b} - p(\mathbf{A})\mathbf{b}\|_2 \leq \min_{\deg(p) < k-1} \|\mathbf{A}\mathbf{A}^{-2}\mathbf{b} - \mathbf{A}p(\mathbf{A})\mathbf{b}\|_2$$
$$\leq \lambda_{\max} \min_{\deg(p) < k-1} \|\mathbf{A}^{-2}\mathbf{b} - p(\mathbf{A})\mathbf{b}\|_2.$$

Plugging the above bounds into (6) and using the fact that the eigenvalues of $\mathbf{T}$ are contained in $[\lambda_{\min}, \lambda_{\max}]$ so that $\|\mathbf{Q}\mathbf{T}^{-1}\mathbf{Q}^{\mathsf{T}}\|_2 \leq \|\mathbf{T}^{-1}\|_2 \leq 1/\lambda_{\min}$,

$$\|\mathbf{A}^{-2}\mathbf{b} - \mathsf{lan}_k(x^{-2}; \mathbf{A}, \mathbf{b})\|_2 \leq 2\kappa(\mathbf{A})^{3/2} \min_{\deg(p) < k-1} \|\mathbf{A}^{-2}\mathbf{b} - p(\mathbf{A})\mathbf{b}\|_2.$$

---

[5]The $\mathbf{A}$-norm is defined for positive definite $\mathbf{A}$ by $\|\mathbf{x}\|_{\mathbf{A}} = \|\mathbf{A}^{1/2}\mathbf{x}\|_2$.

Bounding $\kappa(\mathbf{A})^{3/2}$ by $\kappa(\mathbf{A})^2$ gives the bound in Theorem 4.

Our proof of Theorem 4 generalizes the above approach. We write the Lanczos-FA error for approximating $r(\mathbf{A})\mathbf{b}$ in terms of the error of the optimal approximations to a set of simpler rational functions, and then reduce polynomial approximation of $r(x)$ to polynomial approximation of each of these particular functions.

## 2.2 Implications for non-rational functions

Theorem 4 can be used to derive guarantees for other (non-rational) functions. In particular, consider any function $f(x)$ that is uniformly well approximated by a low-degree rational function $r(x)$ on $\mathcal{I} = [\lambda_{\min}, \lambda_{\max}]$, the interval containing all of the eigenvalues of $\mathbf{A}$; i.e., suppose $\|r - f\|_{\mathcal{I}} :=$ $\max_{x \in \mathcal{I}} |r(x) - f(x)|$ is small. A natural way to approximate $f(\mathbf{A})\mathbf{b}$, used in [4, 5, 59], is to construct $r(x)$ and output $r(\mathbf{A})\mathbf{b}$, using some iterative linear solver to quickly apply the denominators of the partial fraction decomposition of $r(x)$. Alternatively, if we have an instance-optimality guarantee for Lanczos-FA on rational functions like Theorem 4, we could use Lanczos-FA to compute $r(\mathbf{A})\mathbf{b}$. The following analysis shows that simply using Lanczos-FA on $f(x)$ itself cannot be much worse.

**Lemma 5.** *Assume the we have the following instance-optimality guarantee for rational functions:*

$$\|r(\mathbf{A})\mathbf{b} - \mathsf{lan}_k(r; \mathbf{A}, \mathbf{b})\|_2 \le C_r \min_{\deg(p) < c_r k} \|r(\mathbf{A})\mathbf{b} - p(\mathbf{A})\mathbf{b}\|_2. \tag{7}$$

*Here, $C_r$ and $c_r$ depend on the choice of approximant $r(x)$. Then the error of Lanczos-FA on $f(x)$ is bounded as follows:*

$$\|f(\mathbf{A})\mathbf{b} - \mathsf{lan}_k(f; \mathbf{A}, \mathbf{b})\|_2$$
$$\le \min_r \left( (C_r + 2)\|\mathbf{b}\|_2 \cdot \|f - r\|_{\mathcal{I}} + C_r \min_{\deg(p) < c_r k} \|f(\mathbf{A})\mathbf{b} - p(\mathbf{A})\mathbf{b}\|_2 \right). \tag{8}$$

We prove this lemma, which follows directly from the triangle inequality, in Appendix A.5. Compare the form of this bound to that of Definitions 2 and 13. It is close to a near-instance-optimality guarantee, except for the first term, which requires $f(x)$ to be uniformly well-approximated by a rational function $r(x)$ on $[\lambda_{\min}, \lambda_{\max}]$. This is still much stronger than Fact 3, which requires $f(x)$ to be uniformly well-approximated by a *polynomial* to guarantee that Lanczos-FA provides a good approximation. There are many functions with lower degree rational approximations than polynomial approximations, even when we require the rational function $r(x)$ to have poles only in $\mathbb{R} \setminus \mathcal{I}$ (as in our Theorem 4). Such rational approximations are obtainable by the Remez algorithm [64, Chapter 24], and for many important functions they are also known explicitly. For example, a uniform polynomial approximation to the square root on a strictly positive interval $[\lambda_{\min}, \lambda_{\max}]$ requires degree $\Omega(\sqrt{\lambda_{\max}/\lambda_{\min}})$ [64, Chapter 8]. On the other hand, a uniform rational approximation can be obtained with degree only $O(\log(\lambda_{\max}/\lambda_{\min}))$ [37, 64, Chapter 25]. Likewise, a uniform polynomial approximation to $\exp(-x)$ on the interval $[0, B]$ requires degree $\Omega(\sqrt{B})$ [1], but uniform rational approximations can be constructed with no dependence on $B$ [63]. For such functions, we expect (8) to be stronger than Fact 3.

Notice also that, while in Lemma 5, we assume $f(x)$ is well-approximated by a rational function, we are not required to actually construct the approximation. Indeed, since it holds for any $r(x)$, instead of fixing a rational approximation of a certain degree, (8) automatically balances $\|r - f\|_{\mathcal{I}}$, which decreases as the degree grows, with $C_r$, which may increase as the degree grows (see Figure 4).

# 3 Near Spectrum Optimality for $\mathbf{A}^{\pm 1/2}\mathbf{b}$

In the previous section, we proved that Lanczos-FA is nearly instance optimal for rational functions in the sense of Definition 2. In this section, we prove that Lanczos-FA satisfies a weaker form of near optimality for two important non-rational functions: square root and inverse square root. We term this weaker form of guarantee "near spectrum optimality". In Appendix C, we formally define this notion and compare it to Definition 2. We first state our bound for the inverse square root.

**Theorem 6.** *Let $\Lambda$ be the spectrum of $\mathbf{A}$. Then for $k \ge 2$, the Lanczos-FA iterate satisfies the bound*

$$\|\mathbf{A}^{-1/2}\mathbf{b} - \mathsf{lan}_k(x^{-1/2}; \mathbf{A}, \mathbf{b})\|_2 \le \frac{3}{\sqrt{\pi k}} \kappa(\mathbf{A})\|\mathbf{b}\|_2 \min_{\deg(p) < k/2} \left( \max_{x \in \Lambda} \left| \frac{1}{\sqrt{x}} - p(x) \right| \right).$$

That is, Lanczos-FA applied to the inverse square root satisfies Definition 12 ("near spectrum optimality") with

$$C = \frac{3}{\sqrt{\pi k}} \kappa(\mathbf{A}), \qquad c = \frac{1}{2}.$$

We prove this theorem in Appendix D. The proof relies on comparing the error of the $k$th Lanczos iterate for $x^{-1/2}$ to that of the Lanczos iterate for $x^{-1}$. First, applying a bound from [10], we use the Cauchy integral formula to upper bound the error of Lanczos-FA on $x^{\pm 1/2}$ by its error on $x^{-1}$ (Lemma 16). Second, as Equation (17) shows, Lanczos-FA is nearly instance optimal for the function $x^{-1}$; that is, it outputs $p(\mathbf{A})\mathbf{b}$ where $p$ is (nearly) the degree $k$ polynomial that best approximates $x^{-1}$. Third, the best degree $k$ polynomial approximation to $x^{-1}$ must have lower error than the best degree $k/2$ approximation to $x^{-1/2}$. This is because any degree $k/2$ approximation to $x^{-1/2}$ can be squared to yield a good degree $k$ polynomial approximation to $x^{-1}$. Combining these three steps upper bounds the error of the $k$th Lanczos-FA iterate for $x^{-1/2}$ by the error of the best degree $k/2$ polynomial approximation of $x^{-1/2}$.

Nearly the same argument can be used to prove spectrum optimality of Lanczos-FA for the function $\mathbf{A}^{-1/n}\mathbf{b}$ for any $n \in \mathbb{N}$ with $C = (2^n - 1) \cdot \kappa(\mathbf{A})/\sqrt{\pi k}$. Furthermore, using Lemma 15 of Appendix C, we can convert Theorem 6 into a near-instance-optimality guarantee at the price of strong dependence of $C$ on $\mathbf{b}$. We next state our optimality result for the matrix square root.

**Theorem 7.** *Let $\Lambda$ be the spectrum of $\mathbf{A}$. Then for $k \geq 2$, the Lanczos-FA iterate satisfies the bound*

$$\|\mathbf{A}^{1/2}\mathbf{b} - \mathsf{lan}_k(x^{1/2}; \mathbf{A}, \mathbf{b})\| \leq \frac{3\kappa(\mathbf{A})^2}{k^{3/2}} \|\mathbf{b}\|_2 \min_{\deg(p) < k/2+1} \left( \max_{x \in \Lambda \cup \{0\}} \left| \sqrt{x} - p(x) \right| \right).$$

This bound resembles Definition 12 with

$$C = \frac{3\kappa(\mathbf{A})^2}{k^{3/2}}, \qquad c = \frac{1}{2}.$$

However, it is slightly weaker in that the maximization is taken over $\Lambda \cup \{0\}$ instead of only $\Lambda$.

The proof of Theorem 7 is nearly the same as that of Theorem 6, and it likewise appears in Appendix D. Ideally, if $p$ is a polynomial approximation to $x^{1/2}$, we would like to claim that $(p(x)/x)^2$ yields a good polynomial approximation to $x^{-1}$. However, since this function is not necessarily a polynomial, we must instead use use $\left( \frac{p(x)-p(0)}{x} \right)^2$, which introduces the need to include $\{0\}$ in the maximization on the right-hand side.

## 4 Experiments

We now present several numerical experiments to assess the quality of our instance-optimality bounds, Theorem 4 and Lemma 5. Our results show that, despite the large prefactor $C$, our bounds already supersede the standard uniform approximation bound (Fact 3) in many cases. We also compare Lanczos-FA against several recently proposed algorithms for computing matrix functions with strong theoretical guarantees. We find that, in practice, Lanczos-FA performs better than all of them. We implement Lanczos-FA in high precision arithmetic using the `flamp` library, which is based on `mpmath` [51], in order to mitigate any potential impacts of finite precision arithmetic and observe the convergence behavior of the algorithm beyond the standard machine precision.[6]

In Figure 1, we compare the performance of Lanczos to the instance-optimal KSM (which we can compute by direct methods) and against Fact 3 for various matrix functions and spectra. We use three test matrices $\mathbf{A} \in \mathbb{R}^{100 \times 100}$ which all have condition number 100. The first has a uniformly-spaced spectrum, the second has a geometrically-spaced spectrum, and the third has eigenvalues that are all uniformly-spaced in a narrow interval except for ten very small eigenvalues. We compute the bound from Fact 3 using the Remez algorithm and compute the instance-optimal approximation using least squares regression onto the Krylov basis $\mathbf{Q}$. In Figure 1, as in almost all cases we tried, Lanczos-FA

---

[6]Code for our experiments is available at https://github.com/NoahAmsel/lanczos-optimality/tree/neurips2024_near_optimality..

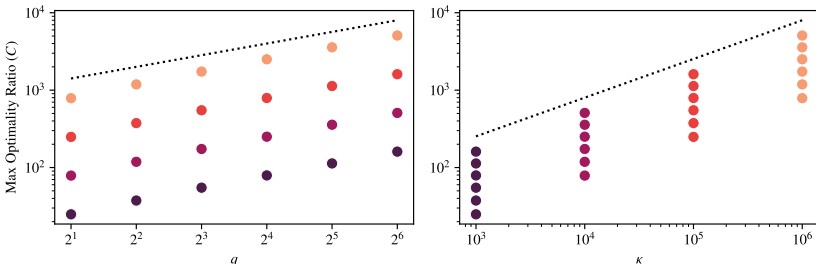

**Figure 3:** The maximum observed ratio between the error of Lanczos-FA and the optimal error over choices of $\mathbf{b}$ when approximating $\mathbf{A}^{-q}$ for matrices with varying condition number $\kappa$. Each point corresponds to a pair $(\kappa, q)$. Points with the same color have the same value of $\kappa$. On the left, the dotted line plots $\sqrt{q\kappa}$ for the maximum $\kappa$ considered ($10^6$). On the right, the dotted line plots $\sqrt{q\kappa}$ for the maximum $q$ considered ($2^6$). Overall, the optimality ratio appears to scale at least as $\Omega(\sqrt{q\kappa})$.

performs nearly as well as the instance-optimal approximation. For instance, the error is never more than a small multiple of the optimal error in the experiments we did.

To better understand Theorem 4, we compare the bound to the true convergence curve of Lanczos-FA for various rational functions in Figure 2. We also plot Fact 3 for reference. We use the same matrices and $\mathbf{b}$ vectors as in Figure 1; results are similar if $\mathbf{b}$ is instead chosen as a uniform linear combination of $\mathbf{A}$'s eigenvectors. We choose rational functions to match the functions used for Figure 1. We construct a degree 5 rational approximation to $\exp(-x/10)$ following [63]. We construct a degree 10 approximation to $\log(x)$ using the BRASIL algorithm [40] and verify that it has real poles outside the interval of the eigenvalues. Despite Theorem 4's exponential dependence on the degree of the rational function being applied, Figure 2 shows that it matches the shape of the convergence curve well and is tighter than Fact 3 when the number of iterations is large. That said, in all cases plotted, Lanczos-FA always returns an approximation much closer to optimal than predicted by Theorem 4, suggesting that the leading coefficient in our bound is pessimistic.

## 4.1 Dependence on the rational function degree

Theorem 4 upper bounds the optimality ratio by $C = q \cdot \kappa(\mathbf{A}_1) \cdot \kappa(\mathbf{A}_2) \cdots \kappa(\mathbf{A}_q)$. We conjecture that this bound is far from tight. However, the following experiment provides evidence that it is not possible to entirely eliminate the dependence on the rational function's denominator degree $q$. In particular, for parameters $(\kappa, q)$, consider approximating $\mathbf{A}^{-q}\mathbf{b}$ where $\mathbf{A}$ has spectrum $\lambda_1 = 1$ and $\lambda_2, \ldots, \lambda_{100}$ evenly spaced between $0.999995 \cdot \kappa$ and $\kappa$. In this case, $\kappa(\mathbf{A}_1) = \cdots = \kappa(\mathbf{A}_q) = \kappa(\mathbf{A}) = \kappa$. We generate a grid of problems by picking different combinations of $(\kappa, q)$ and tuning $\mathbf{b}$ in a limited way to maximize the maximum ratio between the error of Lanczos-FA and the optimal Krylov error over all iterations. In particular, we took $\mathbf{b}$ to be an all ones vector, except we varied its first entry, using grid search to maximize the optimality ratio. The results, plotted in Figure 3, suggest that the optimality ratio grows at least as $\Omega(\sqrt{q \cdot \kappa(\mathbf{A})})$. We have yet to find harder problem instances than this.

## 4.2 Non-rational functions

As noted in Section 2.2, an instance-optimality guarantee for rational functions also implies that Lanczos-FA performs well on functions that are well-approximated by rationals. As an example, we consider the function $f(x) = x^{-0.4}$, for which a rational approximation in any degree can be found using the BRASIL algorithm [40]. Figure 4 shows how applying Lanczos-FA to these rational approximations compares to applying Lanczos-FA directly to the $f(x)$ itself to approximate $\mathbf{A}^{-0.4}\mathbf{b}$. When the number of iterations is small, both methods perform nearly optimally, as the accuracy is limited more by the small size of the Krylov subspace than by the difference between $f(x)$ and the rational approximant (that is, the second term in (8) dominates the first term). As the number of iterations grows, the error due to approximating $f(x)$ in the Krylov subspace continues to decrease while the error of uniformly approximating $f(x)$ by some fixed rational function remains fixed (that is, the first term in (8) dominates the first term); however, increasing the degree of the

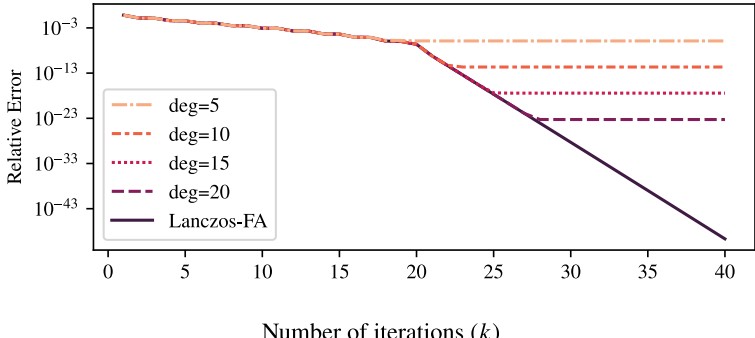

**Figure 4:** Applying Lanczos-FA to the function $\mathbf{A}^{-0.4}$ and rational approximations of various degrees found using the BRASIL algorithm [40]. In this experiment, the spectrum of $\mathbf{A}$ contains two clusters: 10 eigenvalues uniformly spaced near 1, and 90 eigenvalues uniformly spaced near 100. As predicted by the bound in Section 2.2, convergence of Lanczos-FA for this function appears to closely track that of a high degree rational approximant.

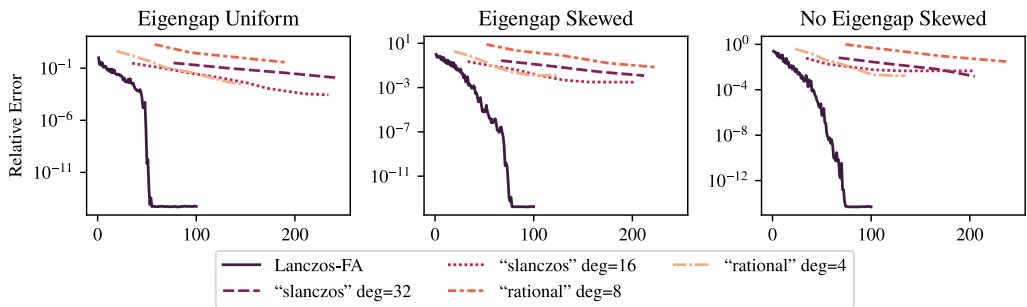

**Figure 5:** A comparison of Lanczos-FA with two methods from [44] ("rational" and "slanczos") for computing the matrix sign function, which work by using a stochastic iterative method to approximate rational approximations to the step function of various degrees. The "rational" method is the main one studied in [44], while "slanczos" is included because it is the best performing in their experiments. Each panel corresponds to one of the test problems from [44]. Iterations of these methods are counted in number of inner products with rows of $\mathbf{A}$ rather than number of matrix-vector products with $\mathbf{A}$ as a whole. To compare these with Lanczos-FA, we consider $d$ such inner products to be equivalent to one matrix-vector product.

rational approximant decreases this source of error. This shows that understanding the convergence of Lanczos-FA for the entire family of rational approximations goes a long way toward explaining its convergence for non-rational functions. In the limit, Lanczos-FA applied to $f(x)$ itself appears to automatically inherit the instance optimality of a suitably high-degree rational approximation.

This result has an additional implication. A number of papers use explicit rational approximations to compute non-rational matrix functions [4, 5, 36, 59]. These approximations are often applied by applying conjugate gradient (or a related method) to each of the terms in the sum [36, 59]. In the case conjugate gradient is used, the resulting algorithm is mathematically identical to Lanczos-FA used to compute the the rational approximation. However, Figure 4 suggests that simply using Lanczos-FA on the original function is both simpler and converges faster (though memory usage and other factors may need to be considered).

Another line of work uses specialized iterative methods that exploit problem structure to apply the rational approximations [29, 2, 53, 44]. In Figure 5, we compare Lanczos-FA to two such methods from [44] for computing $\text{sign}(\mathbf{A})$, for $\mathbf{A}$ of the form $\mathbf{A} = \mathbf{B}^\mathsf{T}\mathbf{B} - \lambda\mathbf{I}$. While they achieve better theoretical bounds than are known for Lanczos-FA, Lanczos-FA far outperforms them on the test problems used in [44].

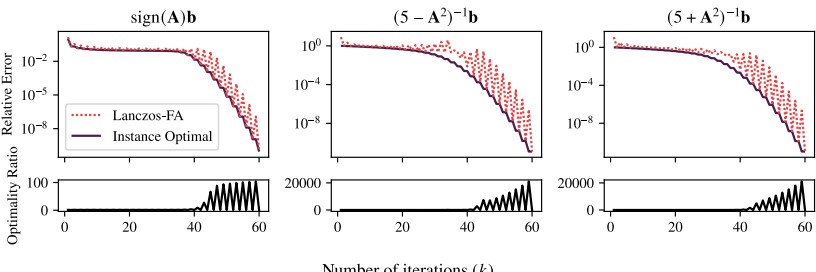

**Figure 6:** Convergence of Lanczos-FA for rational functions with poles in **A**'s eigenvalue range or that are imaginary. The optimality ratio can be very large for some iterations. Similar behavior is seen for functions like sign(**A**) that have a discontinuity in the interval of **A**'s eigenvalues. However, the "overall" convergence of Lanczos-FA still appears to closely track the instance-optimal solution.

**Additional Experiments.** In Section 3 we introduce bounds for the matrix square root and inverse square root, and in Appendix E.1 we provide numerical tests to study the sharpness of the bounds, verifying that they can improve on Fact 3. In Appendix E.2, we demonstrate the convergence behavior of Lanczos-FA on rational functions with poles inside the range of eigenvalues (Figure 6). This illustrates why a bound like Theorem 4 is not possible, but suggests a weaker bound, such as the bound in [12] for $f(x) = 1/x$ and indefinite **A**, may be possible. Appendix E.3 shows that, unlike Lanczos-FA, a related algorithm called Lanczos-OR [11] (which is exactly optimal for rational functions, though not in the Euclidean norm) can perform poorly on high degree rational functions when the error is measured in the Euclidean norm.

## 5 Outlook

This paper provides instance-optimality guarantees for Lanczos-FA applied to a range of rational functions. We conclude with open questions that we believe are worthy of further study.

**Extension to other function classes.** Empirically, Lanczos-FA seems to be nearly instance optimal for a wide variety of functions beyond those considered in this paper, such as rational functions with conjugate pairs of *complex* poles whose real parts lie outside $[\lambda_{\min}, \lambda_{\max}]$. As seen in Figure 6, the error of Lanczos-FA on functions with real poles in $[\lambda_{\min}, \lambda_{\max}]$ is intriguing, oscillating between being very large and nearly optimal. We discuss this more in Appendix E.2. It would be valuable to provide bounds explaining these behaviors.

It would be also natural to try to extend our bounds to Stieltjes/Markov functions, which can be viewed as a certain type of infinite degree rational function approximations with poles in $(-\infty, 0]$, and includes important functions like the inverse square root and a shifted logartihm. Our bound Theorem 4 cannot be directly applied to this class due to the dependence on the rational function denominator degree $q$.

**Construction of hard instances / refined upper bounds.** Theorem 4 has an exponential dependence on the degree of the rational function's denominator $q$, which limits the practicality of our bounds. It is unclear if and when this dependence can be improved. The experiment in Section 4.1 provides strong evidence that some dependence on $q$ is necessary, but the hardest examples we have depend on $\sqrt{q}$, instead of the current bound of $\kappa(\mathbf{A})^q$ guaranteed by Theorem 4. It is an open question whether Theorem 4 can be tightened, or whether matching hard instances exist.

**Finite precision arithmetic.** Our analysis concerns the behavior of the Lanczos algorithm when run in exact arithmetic. In practice, the implementation of the Lanczos algorithm is very important; for instance, practical implementations often output a **Q** which is far from orthogonal [50, 8]. While this instability can be mitigated with more expensive implementations, theoretical work shows that, surprisingly, Lanczos and Lanczos-FA can work well despite it [57, 56, 55, 33, 21, 20]. For example, [21, 52, 9] show that Fact 3 still holds up to a close approximation in finite precision arithmetic for any bounded matrix function. It would be valuable to study whether stronger near-optimality guarantees like those proven in Theorem 4 are also robust to finite precision. For $f(x) = 1/x$, this problem has been studied in [33].

**Acknowledgments:** This research was supported in part by NSF Awards 2045590 (Chen, Ch. Musco), 2046235 (Ca. Musco), 2427363 (Chen, Ca. Musco, Ch. Musco), and 2234660 (Amsel).

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

# A  Proof of Theorem 4

## A.1  Notation

We first introduce notation used throughout. Given a rational function $r(x) = n(x)/m(x)$ with numerator degree $p$ denominator degree $q$, for $1 \le i \le j \le q$, we define:

$$m_{i,j}(x) := \prod_{k=i}^{j} (x - z_k).$$

We adopt the convention that $m_{j+1,j}(x) := 1$. Note that $m_{i,j}$ is a $(j - i + 1)$-degree polynomial. Define also

$$r_j(x) := n(x)/m_{1,j}(x).$$

Note that $m_{1,q} = m$, so $r_q = r(x)$. Recall that for any function $f(x)$, the Lanczos-FA approximation to $f(\mathbf{A})\mathbf{b}$ is $\mathsf{lan}_k(f; \mathbf{A}, \mathbf{b}) = \mathbf{Q}f(\mathbf{T})\mathbf{Q}^\mathsf{T}\mathbf{b}$. Define

$$\mathsf{err}_k(f) := f(\mathbf{A})\mathbf{b} - \mathsf{lan}_k(f; \mathbf{A}, \mathbf{b}).$$

Proving Theorem 4 amounts to proving an upper bound on $\|\mathsf{err}_k(r)\|_2$. Finally, for any symmetric positive definite matrix $\mathbf{M}$, define

$$\mathsf{opt}_k(f, \mathbf{M}; \mathbf{A}, \mathbf{b}) := \operatorname*{argmin}_{\mathbf{x} \in \mathcal{K}_k(\mathbf{A}, \mathbf{b})} \|f(\mathbf{A})\mathbf{b} - \mathbf{x}\|_\mathbf{M}.$$

For brevity, in the analysis below we will write $\mathsf{lan}_k(f)$ and $\mathsf{opt}_k(f, \mathbf{M})$ in place of $\mathsf{lan}_k(f; \mathbf{A}, \mathbf{b})$ and $\mathsf{opt}_k(f, \mathbf{M}; \mathbf{A}, \mathbf{b})$, since $\mathbf{A}$ and $\mathbf{b}$ are fixed throughout the analysis.

## A.2  Simplifying $\mathsf{lan}_k(r_j)$ and $\mathsf{opt}_k(r_j, \mathbf{A}_j)$

We begin with a few standard results that will be useful for the proof of Theorem 4.

**Lemma 8** ([22, 61])**.** *For a polynomial $n(x)$, $\mathsf{lan}_k(n) = n(\mathbf{A})\mathbf{b}$ if $k > \deg(n)$.*

*Proof.* By definition of the Krylov subspace, $\mathbf{A}^j\mathbf{b} \in \mathcal{K}_k(\mathbf{A}, \mathbf{b})$ for all $j < k$. Since $\mathbf{Q}\mathbf{Q}^\mathsf{T}$ is the 2-norm orthogonal projector onto $\mathcal{K}_k(\mathbf{A}, \mathbf{b})$, then $\mathbf{Q}\mathbf{Q}^\mathsf{T}\mathbf{A}^j\mathbf{b} = \mathbf{A}^j\mathbf{b}$ for all $j < k$. Iteratively applying this fact,

$$\mathbf{A}^j\mathbf{b} = \mathbf{Q}\mathbf{Q}^\mathsf{T}\mathbf{A}^j\mathbf{b} = \mathbf{Q}\mathbf{Q}^\mathsf{T}\mathbf{A}\mathbf{A}^{j-1}\mathbf{b} = \mathbf{Q}\mathbf{Q}^\mathsf{T}\mathbf{A}\mathbf{Q}\mathbf{Q}^\mathsf{T}\mathbf{A}^{j-1}\mathbf{b} = \mathbf{Q}\mathbf{Q}^\mathsf{T}\mathbf{A}\mathbf{Q} \cdots \mathbf{Q}^\mathsf{T}\mathbf{A}\mathbf{Q}\mathbf{Q}^\mathsf{T}\mathbf{b}.$$

Using that $\mathbf{Q}^\mathsf{T}\mathbf{A}\mathbf{Q} = \mathbf{T}$, we find that $\mathbf{A}^j\mathbf{b} = \mathbf{Q}\mathbf{T}^j\mathbf{Q}^\mathsf{T}\mathbf{b}$, and thus, by linearity, $n(\mathbf{A})\mathbf{b} = \mathbf{Q}n(\mathbf{T})\mathbf{Q}^\mathsf{T}\mathbf{b} = \mathsf{lan}_k(n)$. $\qquad\square$

**Lemma 9.** *Let $\mathbf{A}_j$ be as in (5). Then, for any $k \ge 0$, $\mathsf{opt}_k(r_j, \mathbf{A}_j) = \mathbf{Q}(\mathbf{T} - z_j\mathbf{I})^{-1}\mathbf{Q}^\mathsf{T}r_{j-1}(\mathbf{A})\mathbf{b}$.*

*Proof.* The $\mathbf{A}_j$-norm projector onto $\mathcal{K}_k(\mathbf{A}, \mathbf{b})$ is $\mathbf{Q}(\mathbf{Q}^\mathsf{T}\mathbf{A}_j\mathbf{Q})^{-1}\mathbf{Q}^\mathsf{T}\mathbf{A}_j$, so

$$\mathsf{opt}_k(r_j, \mathbf{A}_j) = \mathbf{Q}(\mathbf{Q}^\mathsf{T}\mathbf{A}_j\mathbf{Q})^{-1}\mathbf{Q}^\mathsf{T}\mathbf{A}_jr_j(\mathbf{A}). \tag{9}$$

Recall that $\mathbf{A}_j = \pm(\mathbf{A} - z_j\mathbf{I})$. Since $\mathbf{Q}^\mathsf{T}\mathbf{A}\mathbf{Q} = \mathbf{T}$ and $\mathbf{Q}^\mathsf{T}\mathbf{Q} = \mathbf{I}$, we have $\mathbf{Q}^\mathsf{T}\mathbf{A}_j\mathbf{Q} = \pm(\mathbf{Q}^\mathsf{T}(\mathbf{A} - z_j\mathbf{I})\mathbf{Q}) = \pm(\mathbf{T} - z_j\mathbf{I})$. Noting also that $\mathbf{A}_jr_j(\mathbf{A}) = \pm r_{j-1}(\mathbf{A})$ and plugging into (9) gives the result. $\qquad\square$

## A.3  A telescoping sum for the error in terms of $\mathsf{opt}_k(r_j, \mathbf{A}_j)$

Our first main result is a decomposition for the Lanczos-FA error.

**Lemma 10.** *Suppose $k > \deg(n)$. Then,*

$$r(\mathbf{A})\mathbf{b} - \mathsf{lan}_k(r) = \sum_{j=1}^{q} \left[ \prod_{i=j+1}^{q} \mathbf{Q}(\mathbf{T} - z_i\mathbf{I})^{-1}\mathbf{Q}^\mathsf{T} \right] (r_j(\mathbf{A})\mathbf{b} - \mathsf{opt}_k(r_j, \mathbf{A}_j)).$$

*where we adopt the convention that $\prod_{i=q+1}^{q} \mathbf{B}_i = \mathbf{I}$ for any set of matrices $\{\mathbf{B}_i\}$.*

*Proof.* We can decompose $\text{err}_k(r_j)$ as

$$\text{err}_k(r_j) = r_j(\mathbf{A})\mathbf{b} - \mathbf{Q}r_j(\mathbf{T})\mathbf{Q}^\mathsf{T}\mathbf{b}$$
$$= \left[ r_j(\mathbf{A})\mathbf{b} - \text{opt}_k(r_j, \mathbf{A}_j) \right] + \left[ \text{opt}_k(r_j, \mathbf{A}_j) - \mathbf{Q}r_j(\mathbf{T})\mathbf{Q}^\mathsf{T}\mathbf{b} \right]. \tag{10}$$

Focusing on the second term and using [Lemma 9](),

$$\text{opt}_k(r_j, \mathbf{A}_j) - \mathbf{Q}r_j(\mathbf{T})\mathbf{Q}^\mathsf{T}\mathbf{b}$$
$$= \mathbf{Q}(\mathbf{T} - z_j\mathbf{I})^{-1}\mathbf{Q}^\mathsf{T}r_{j-1}(\mathbf{A})\mathbf{b} - \mathbf{Q}(\mathbf{T} - z_j\mathbf{I})^{-1}r_{j-1}(\mathbf{T})\mathbf{Q}^\mathsf{T}\mathbf{b}$$
$$= \mathbf{Q}(\mathbf{T} - z_j\mathbf{I})^{-1}\mathbf{Q}^\mathsf{T}\left[ r_{j-1}(\mathbf{A})\mathbf{b} - \mathbf{Q}r_{j-1}(\mathbf{T})\mathbf{Q}^\mathsf{T}\mathbf{b} \right]$$
$$= \mathbf{Q}(\mathbf{T} - z_j\mathbf{I})^{-1}\mathbf{Q}^\mathsf{T}\left[ \text{err}_k(r_{j-1}) \right].$$

Substituting this into [(10)](), we have

$$\text{err}_k(r_j) = \left[ r_j(\mathbf{A})\mathbf{b} - \text{opt}_k(r_j, \mathbf{A}_j) \right] + \mathbf{Q}(\mathbf{T} - z_j\mathbf{I})^{-1}\mathbf{Q}^\mathsf{T}\text{err}_k(r_{j-1}). \tag{11}$$

Next, notice that $\text{opt}_k(r_1, \mathbf{A}_1)$ is exactly the Lanczos approximation to $r_1(\mathbf{A})\mathbf{b}$. Indeed, since $\mathbf{Q}^\mathsf{T}\mathbf{Q} = \mathbf{I}$, [Lemmas 8]() and [9]() imply

$$\text{lan}_k(r_1) = \mathbf{Q}(\mathbf{T} - z_i\mathbf{I})^{-1}n(\mathbf{T})\mathbf{Q}^\mathsf{T}\mathbf{b}$$
$$= \mathbf{Q}(\mathbf{T} - z_i\mathbf{I})^{-1}\mathbf{Q}^\mathsf{T}\mathbf{Q}n(\mathbf{T})\mathbf{Q}^\mathsf{T}\mathbf{b}$$
$$= \mathbf{Q}(\mathbf{T} - z_1\mathbf{I})^{-1}\mathbf{Q}^\mathsf{T}n(\mathbf{A})\mathbf{b} = \text{opt}_k(r_1, \mathbf{A}_1).$$

Using this as a base case, we can repeatedly apply our decomposition of $\text{err}_k(r_j)$ in [(11)]() to obtain

$$\text{err}_k(r) = \text{err}_k(r_q) = \sum_{j=1}^{q} \left[ \prod_{i=j+1}^{q} \mathbf{Q}(\mathbf{T} - z_i\mathbf{I})^{-1}\mathbf{Q}^\mathsf{T} \right] \left( r_j(\mathbf{A})\mathbf{b} - \text{opt}_k(r_j, \mathbf{A}_j) \right).$$

$\square$

### A.4    Bounding each term in the telescoping sum and combining

The following lemma allows us to relate the optimality of the functions $r_j$ as measured in the $\mathbf{A}_j$-norm to that of $r_q = r$ as measured in the 2-norm.

**Lemma 11.** *For any $k > q - j$,*
$$\|r_j(\mathbf{A})\mathbf{b} - \text{opt}_k(r_j, \mathbf{A}_j)\|$$
$$\leq \kappa(\mathbf{A}_j)^{1/2} \cdot \|m_{j+1,q}(\mathbf{A})\|_2 \min_{\deg(p)<k-(q-j)} \|r(\mathbf{A})\mathbf{b} - p(\mathbf{A})\mathbf{b}\|_2.$$

*Proof.* Recall that $\text{opt}_k(r_j, \mathbf{A}_j)$ is the optimal approximation to $r_j(\mathbf{A})\mathbf{b}$ in the $\mathbf{A}_j$-norm, so can be related to the optimal approximation in the 2-norm.

$$\|r_j(\mathbf{A})\mathbf{b} - \text{opt}_k(r_j, \mathbf{A}_j)\|_2 = \left\| \mathbf{A}_j^{-1/2}\mathbf{A}_j^{1/2}(r_j(\mathbf{A})\mathbf{b} - \text{opt}_k(r_j, \mathbf{A}_j)) \right\|_2$$
$$\leq \|\mathbf{A}_j^{-1/2}\|_2 \cdot \|r_j(\mathbf{A})\mathbf{b} - \text{opt}_k(r_j, \mathbf{A}_j)\|_{\mathbf{A}_j}$$
$$= \|\mathbf{A}_j^{-1}\|_2^{1/2} \min_{\deg(p)<k} \|r_j(\mathbf{A})\mathbf{b} - p(\mathbf{A})\mathbf{b}\|_{\mathbf{A}_j}$$
$$\leq \|\mathbf{A}_j^{-1}\|_2^{1/2} \cdot \|\mathbf{A}_j\|_2^{1/2} \min_{\deg(p)<k} \|r_j(\mathbf{A})\mathbf{b} - p(\mathbf{A})\mathbf{b}\|_2$$
$$= \kappa(\mathbf{A}_j)^{1/2} \min_{\deg(p)<k} \|r_j(\mathbf{A})\mathbf{b} - p(\mathbf{A})\mathbf{b}\|_2.$$

We now relate the error of approximating $r_j(\mathbf{A})\mathbf{b}$ to that of approximating $r_q(\mathbf{A})\mathbf{b} = r(\mathbf{A})\mathbf{b}$:

$$\min_{\deg(p)<k} \|r_j(\mathbf{A})\mathbf{b} - p(\mathbf{A})\mathbf{b}\|_2$$
$$= \min_{\deg(p)<k} \|m_{j+1,q}(\mathbf{A})r_q(\mathbf{A})\mathbf{b} - p(\mathbf{A})\mathbf{b}\|_2$$
$$\leq \min_{\deg(p)<k-(q-j)} \|m_{j+1,q}(\mathbf{A})r_q(\mathbf{A})\mathbf{b} - m_{j+1,q}(\mathbf{A})p(\mathbf{A})\mathbf{b}\|_2$$
$$\leq \|m_{j+1,q}(\mathbf{A})\|_2 \min_{\deg(p)<k-(q-j)} \|r(\mathbf{A})\mathbf{b} - p(\mathbf{A})\mathbf{b}\|_2.$$

Combining these two steps proves the lemma. □

With the above results in place, we can prove Theorem 4. We will apply the triangle inequality to the telescoping sum for $\text{err}_k(r)$ (Lemma 10) and bound each term by the error of the optimal approximation to $r(\mathbf{A})\mathbf{b}$ (Lemma 11).

*Proof of Theorem 4.* Focus on a single term in the sum in Lemma 10. First we will get rid of $\mathbf{Q}$ and $\mathbf{Q}^\mathsf{T}$. For $1 \le j < q$, using that $\mathbf{Q}^\mathsf{T}\mathbf{Q} = \mathbf{I}$ we have that

$$\left\|\prod_{i=j+1}^{q} \mathbf{Q}(\mathbf{T} - z_i\mathbf{I})^{-1}\mathbf{Q}^\mathsf{T}\right\|_2 = \|\mathbf{Q}m_{j+1,q}(\mathbf{T})^{-1}\mathbf{Q}^\mathsf{T}\|_2 \le \|m_{j+1,q}(\mathbf{T})^{-1}\|_2. \tag{12}$$

Also note that by the convention adopted in Lemma 10, for $j = q$, $\prod_{i=j+1}^{q} \mathbf{Q}(\mathbf{T} - z_k\mathbf{I})^{-1}\mathbf{Q}^\mathsf{T} = \mathbf{I} = m_{j+1,q}(\mathbf{T})$. Next we claim that

$$\|m_{j+1,q}(\mathbf{T})^{-1}\|_2 \le \|m_{j+1,q}(\mathbf{A})^{-1}\|_2. \tag{13}$$

To see why this is the case, note that because $\mathbf{T} = \mathbf{Q}^\mathsf{T}\mathbf{A}\mathbf{Q}$, the eigenvalues of $\mathbf{T}$ are contained in $\mathcal{I} = [\lambda_{\min}(\mathbf{A}), \lambda_{\max}(\mathbf{A})]$. By assumption, the roots of $m_{j+1,q}(x)$ are all real and lie outside of $\mathcal{I}$. Since there is at most one critical point between distinct roots, there can only be one critical point of $m_{j+1,q}(x)$ in $\mathcal{I}$. Thus, there can be no local minima of $|m_{j+1,q}(x)|$ in the interior of $\mathcal{I}$. Rather, the minimum of $|m_{j+1,q}(x)|$ over $\mathcal{I}$ must be attained at the boundary; i.e. at $x = \lambda_{\min}$ or $x = \lambda_{\max}$.

We now apply the triangle inequality to the telescoping sum of Lemma 10:

$$\|\text{err}_k(r)\|_2 \le \sum_{j=1}^{q} \left\|\prod_{i=j+1}^{q} \mathbf{Q}(\mathbf{T} - z_i\mathbf{I})^{-1}\mathbf{Q}^\mathsf{T}\right\|_2 \cdot \left\|r_j(\mathbf{A})\mathbf{b} - \text{opt}_k(r_j, \mathbf{A}_j)\right\|_2.$$

Applying (12) and (13) we then have

$$\|\text{err}_k(r)\|_2 \le \sum_{j=1}^{q} \|m_{j+1,q}(\mathbf{A})^{-1}\|_2 \cdot \left\|r_j(\mathbf{A})\mathbf{b} - \text{opt}_k(r_j, \mathbf{A}_j)\right\|_2.$$

Finally, using Lemma 11 we find

$$\|\text{err}_k(r)\|_2 \le \sum_{j=1}^{q} \kappa(\mathbf{A}_j)^{1/2} \cdot \kappa(m_{j+1,q}(\mathbf{A})) \min_{\deg(p)<k-(q-j)} \|p(\mathbf{A})\mathbf{b} - r(\mathbf{A})\mathbf{b}\|_2. \tag{14}$$

We can simplify by noting that

$$\min_{\deg(p)<k-(q-j)} \|p(\mathbf{A})\mathbf{b} - r(\mathbf{A})\mathbf{b}\|_2 \le \min_{\deg(p)<k-(q-1)} \|p(\mathbf{A})\mathbf{b} - r(\mathbf{A})\mathbf{b}\|_2,$$

and we can combine all the condition number factors using

$$\sum_{j=1}^{q} \kappa(\mathbf{A}_j)^{1/2} \cdot \kappa(m_{j+1,q}(\mathbf{A})) \le \sum_{j=1}^{q} \kappa(\mathbf{A}_j) \prod_{i=j+1}^{q} \kappa(\mathbf{A}_i) \le q \prod_{i=1}^{q} \kappa(\mathbf{A}_i).$$

The result follows by plugging into (14). □

## A.5 Proof of Lemma 5

We can bound the error of Lanczos-FA on $f(x)$ using triangle inequality as follows:

$$\|f(\mathbf{A})\mathbf{b} - \text{lan}_k(f; \mathbf{A}, \mathbf{b})\|_2 \le \|f(\mathbf{A})\mathbf{b} - r(\mathbf{A})\mathbf{b}\|_2 + \|r(\mathbf{A})\mathbf{b} - \text{lan}_k(r; \mathbf{A}, \mathbf{b})\|_2$$
$$+ \|\text{lan}_k(r; \mathbf{A}, \mathbf{b}) - \text{lan}_k(f; \mathbf{A}, \mathbf{b})\|_2. \tag{15}$$

The first and third terms of (15) are controlled by the maximum error of approximating $f(x)$ with $r(x)$ over $\mathcal{I}$. Specifically, if we let $\|r - f\|_{\mathcal{I}} := \max_{x \in \mathcal{I}} |r(x) - f(x)|$,

$$\|f(\mathbf{A})\mathbf{b} - r(\mathbf{A})\mathbf{b}\|_2 \le \|\mathbf{b}\|_2 \cdot \|f(\mathbf{A}) - r(\mathbf{A})\|_2 \le \|\mathbf{b}\|_2 \cdot \|r - f\|_{\mathcal{I}}$$

and similarly, using that $\Lambda(\mathbf{T}) \subset \mathcal{I}$,

$$\|\mathsf{lan}_k(r; \mathbf{A}, \mathbf{b}) - \mathsf{lan}_k(f; \mathbf{A}, \mathbf{b})\|_2 = \|\mathbf{Q}r(\mathbf{T})\mathbf{Q}^\mathsf{T}\mathbf{b} - \mathbf{Q}f(\mathbf{T})\mathbf{Q}^\mathsf{T}\mathbf{b}\|_2 \le \|\mathbf{b}\|_2 \cdot \|r - f\|_{\mathcal{I}}.$$

The second term of (15) can controlled using (7) and a triangle inequality:

$$\begin{aligned}
\|r(\mathbf{A})\mathbf{b} - \mathsf{lan}_k(r; \mathbf{A}, \mathbf{b})\|_2 &\le C_r \min_{\deg(p) < c_r k} \|r(\mathbf{A})\mathbf{b} - p(\mathbf{A})\mathbf{b}\|_2 \\
&\le C_r \min_{\deg(p) < c_r k} (\|r(\mathbf{A})\mathbf{b} - f(\mathbf{A})\mathbf{b}\|_2 + \|f(\mathbf{A})\mathbf{b} - p(\mathbf{A})\mathbf{b}\|_2) \\
&\le C_r \|\mathbf{b}\|_2 \cdot \|r - f\|_{\mathcal{I}} + C_r \min_{\deg(p) < c_r k} \|f(\mathbf{A})b - p(\mathbf{A})\mathbf{b}\|_2.
\end{aligned}$$

Combining, we obtain the bound

$$\begin{aligned}
&\|f(\mathbf{A})\mathbf{b} - \mathsf{lan}_k(f; \mathbf{A}, \mathbf{b})\|_2 \\
&\quad \le \min_r \left( (C_r + 2)\|\mathbf{b}\|_2 \cdot \|r - f\|_{\mathcal{I}} + C_r \min_{\deg(p) < c_r k} \|f(\mathbf{A})\mathbf{b} - p(\mathbf{A})\mathbf{b}\|_2 \right). \quad (16)
\end{aligned}$$

# B  Comparison to Prior Work

## B.1  Details of existing near-optimality guarantees for Lanczos-FA

In this section, we review in detail the prior analyses of Lanczos-FA cited in Section 1.3. These are the only near-optimality guarantees for Lanczos-FA of which we are aware. Instance optimality trivially holds when $f(x)$ is a polynomial of degree $< k$. In this case, it is well known that $\mathsf{lan}_k(f; \mathbf{A}, \mathbf{b})$ exactly applies $f(\mathbf{A})\mathbf{b}$; i.e., $\|f(\mathbf{A})\mathbf{b} - \mathsf{lan}_k(f; \mathbf{A}, \mathbf{b})\|_2 = 0$ [61]. When $f(x) = 1/x$ and $\mathbf{A}$ is positive definite, the Lanczos-FA algorithm is mathematically equivalent to the well-known conjugate gradient algorithm for solving a system $\mathbf{Ax} = \mathbf{b}$. This implies that Lanczos-FA is the optimal approximation in the Krylov subspace with respect to the $\mathbf{A}$-norm; that is,

$$\mathsf{lan}_k(1/x; \mathbf{A}, \mathbf{b}) = \tilde{p}(\mathbf{A})\mathbf{b}, \qquad \tilde{p} := \underset{\deg(p) < k}{\operatorname{argmin}} \|\mathbf{A}^{-1}\mathbf{b} - p(\mathbf{A})\mathbf{b}\|_{\mathbf{A}}.$$

This immediately yields near instance optimality in the Euclidean norm:

$$\|\mathbf{A}^{-1}\mathbf{b} - \mathsf{lan}_k(1/x; \mathbf{A}, \mathbf{b})\|_2 \le \sqrt{\kappa(\mathbf{A})} \min_{\deg(p) < k} \|\mathbf{A}^{-1}\mathbf{b} - p(\mathbf{A})\mathbf{b}\|_2, \qquad (17)$$

where $\kappa(\mathbf{A})$ is the condition number $\lambda_{\max}/\lambda_{\min}$ of $\mathbf{A}$. That is, Definition 2 is satisfied with $C = \sqrt{\kappa(\mathbf{A})}$ and $c = 1$.

Note that the best polynomial approximation to $1/x$ on $\mathcal{I} = [\lambda_{\min}, \lambda_{\max}]$ already converges at a geometric rate [45]:

$$\min_{\deg(p) < k} \max_{x \in \mathcal{I}} |1/x - p(x)| = \frac{8t^{k+1}}{(t^2 - 1)^2(\lambda_{\max} - \lambda_{\min})}, \qquad t := 1 - \frac{2}{1 + \sqrt{\kappa(\mathbf{A})}}.$$

That is, Fact 3 suffices to prove the exponential convergence of the conjugate gradient method. However, conjugate gradient (and equivalently Lanczos-FA) often converges even faster, which is explained theoretically by the stronger instance- and spectrum-optimality guarantees that the method satisfies [6, 47, 8].

If $\mathbf{A}$ is not positive definite, $\mathbf{T}$ may have an eigenvalue at or near to zero and the error of the Lanczos-FA approximation to $\mathbf{A}^{-1}\mathbf{b}$ can be arbitrarily large. In fact, the same is true for any function $f(x)$ which is much larger on $\mathcal{I} = [\lambda_{\min}, \lambda_{\max}]$ than on $\Lambda$, the set of eigenvalues of $\mathbf{A}$. However, while the Lanczos-FA iterates may be bad at some iterations, the overall convergence of Lanczos-FA with $f(x) = 1/x$ is actually good [12]. In particular, the convergence of Lanczos-FA can be related to the MINRES algorithm [15, 49], which produces an optimal approximation to $\mathbf{A}^{-1}\mathbf{b}$ with respect to the $\mathbf{A}^2$-norm. This allows certain optimality guarantees for MINRES to be transferred to Lanczos-FA, even on indefinite systems. In particular, [12] asserts that for every $k$, there exists $k^* \le k$ such that[7]

$$\|\mathbf{A}^{-1}\mathbf{b} - \mathsf{lan}_{k^*}(1/x; \mathbf{A}, \mathbf{b})\|_2 \le \kappa(\mathbf{A})\sqrt{k + 1} \min_{\deg(p) < k} \|\mathbf{A}^{-1}\mathbf{b} - p(\mathbf{A})\mathbf{b}\|_2. \qquad (18)$$

---

[7]For indefinite matrices, $\kappa(\mathbf{A})$ denotes the ratio of the largest eigenvalue magnitude to the smallest eigenvalue magnitude.

While (18) does not quite fit the form of Definition 2 because of the $k^*$ on the left side, it is similar in spirit. Also note that the dependence on $k$ in the prefactor $\sqrt{k+1}$ is not of great significance. Indeed, the convergence of the optimal polynomial approximation to $1/x$ on two intervals bounded away from zero is geometric.

Finally, a guarantee for the matrix exponential is proved in [20, (45)]. They show that the Lanczos-FA iterate satisfies the guarantee

$$\| \exp(-t\mathbf{A})\mathbf{b} - \mathsf{lan}_k(\exp(-tx); \mathbf{A}, \mathbf{b}) \|_2$$
$$\leq 3\|\mathbf{A}\|_2^2 \, t^2 \max_{0 \leq s \leq t} \left( \min_{\deg(p) < k-2} \| \exp(-s\mathbf{A})\mathbf{b} - p(\mathbf{A})\mathbf{b} \|_2 \right).$$

Again, this bound does not quite fit Definition 2 due to the maximization over $s$. The authors of [20] state, "It is not known whether the maximum over $s \in [0, t]$ in (45) can be omitted and $s$ set equal to $t$ in the right-hand side."

### B.2 Comparison to Lanczos-OR

In [11] an algorithm called the Lanczos method for optimal rational function approximation (Lanczos-OR) was developed. For rational functions of the form (4), when run for $k > \deg(n)$ iterations, Lanczos-OR produces iterates

$$\mathsf{lanOR}_k(r; \mathbf{A}, \mathbf{b}) := \underset{\mathbf{x} \in \mathcal{K}_{k-\lfloor q/2 \rfloor}}{\mathrm{argmin}} \|r(\mathbf{A})\mathbf{b} - \mathbf{x}\|_{|r(\mathbf{A})|}.$$

This implies a 2-norm instance-optimality guarantee (Definition 2):

$$\|r(\mathbf{A})\mathbf{b} - \mathsf{lanOR}_k(r; \mathbf{A}, \mathbf{b})\|_2 \leq \sqrt{\kappa(r(\mathbf{A}))} \min_{\deg(p) < k-\lfloor q/2 \rfloor} \|r(\mathbf{A})\mathbf{b} - p(\mathbf{A})\mathbf{b}\|_2. \qquad (19)$$

This guarantee is similar to and often somewhat better than Theorem 4. For example, when $r(x) = 1/m(x)$, $\sqrt{\kappa(r(\mathbf{A}))} = \sqrt{\kappa(\mathbf{A}_1) \cdot \kappa(\mathbf{A}_2) \cdots \kappa(\mathbf{A}_q)}$. In this case (19) improves on Theorem 4 by a $q$ factor and a square root. However, the bound is for a different algorithm; it does not extend to the ubiquitous Lanczos-FA method.

Moreover, despite this bound, Lanczos-OR usually performs worse than Lanczos-FA in practice if error is measured in the 2-norm. It is not hard to find examples where (19) is essentially sharp; see for instance Figure 8. On the other hand, we have been unable to find any numerical examples where Theorem 4 is sharp, suggesting that it may be possible to prove a tighter bound for Lanczos-FA.

## C  Instance, spectrum, and FOV optimality

The convergence of Lanczos-FA is entirely determined by spectral properties of $\mathbf{A}$. Therefore, following the classical analysis of $f(x) = 1/x$, we can relax the notion of instance optimality (Definition 2) by removing its dependence on $\mathbf{b}$:

**Definition 12** (Near Spectrum Optimality). *For an input instance $(f, \mathbf{A}, \mathbf{b}, k)$, we say that a Krylov method is nearly spectrum optimal with parameters $C, c$ if*

$$\|f(\mathbf{A})\mathbf{b} - \mathsf{alg}_k(f; \mathbf{A}, \mathbf{b})\|_2 \leq C\|\mathbf{b}\|_2 \min_{\deg(p) < ck} \|f(\mathbf{A}) - p(\mathbf{A})\|_2.$$

Note that Definition 12 depends only on the spectrum of $\mathbf{A}$, and not on the interaction of the eigenvectors of $\mathbf{A}$ with the starting vector $\mathbf{b}$. Hence, we call it "spectrum optimality". Note also that the guarantee of Theorem 6 fits this form. We can further relax this notion of optimality by weakening its dependence on $\mathbf{A}$:

**Definition 13** (Near FOV Optimality). *For an input instance $(f, \mathbf{A}, \mathbf{b}, k)$, we say that a Krylov method is nearly FOV optimal with parameters $C, c$ if*

$$\|f(\mathbf{A})\mathbf{b} - \mathsf{alg}_k(f; \mathbf{A}, \mathbf{b})\|_2 \leq C\|\mathbf{b}\|_2 \min_{\deg(p) < ck} \left( \max_{x \in [\lambda_{\min}, \lambda_{\max}]} |f(x) - p(x)| \right).$$

Note that Fact 3 fits this form. Each type of optimality tightly relaxes the previous one. Definition 2 implies Definition 12 and, as we show in Theorem 14, for any $f$, $k$ and $\mathbf{A}$, there exists a "worst-case" choice of $\mathbf{b}$ for which their bounds coincide; likewise, Definition 12 implies Definition 13, and for any $f(x)$ and $k$ there exists a "worst-case" choice of $\mathbf{A}$ and $\mathbf{b}$ for which their bounds coincide.

**Theorem 14.**

1. *Given any $d$, discrete set $\Lambda$ of at most $d$ real points, $f(x)$, and $k < d$, there exists a symmetric matrix $\mathbf{A} \in \mathbb{R}^{d \times d}$ and vector $\mathbf{b} \in \mathbb{R}^d$ such that the spectrum of $\mathbf{A}$ is $\Lambda$ and*

$$\min_{\deg(p)<k} \|f(\mathbf{A})\mathbf{b} - p(\mathbf{A})\mathbf{b}\|_2 / \|\mathbf{b}\|_2 = \min_{\deg(p)<k} \max_{x \in \Lambda} |f(x) - p(x)|.$$

2. *Given any $d$, $\lambda_{\min}$, $\lambda_{\max}$, $f(x)$ continuous on $[\lambda_{\min}, \lambda_{\max}]$, and $k < d$, there exists a symmetric matrix $\mathbf{A} \in \mathbb{R}^{d \times d}$ and vector $\mathbf{b} \in \mathbb{R}^d$ such that the smallest eigenvalue of $\mathbf{A}$ is $\lambda_{\min}$, the largest eigenvalue of $\mathbf{A}$ is $\lambda_{\max}$, and*

$$\min_{\deg(p)<k} \|f(\mathbf{A})\mathbf{b} - p(\mathbf{A})\mathbf{b}\|_2 / \|\mathbf{b}\|_2 = \min_{\deg(p)<k} \max_{x \in \mathcal{I}} |f(x) - p(x)|.$$

Our proof follows the approach of [34, Theorem 1] closely.

*Proof.* For Item 1, note that if $|\Lambda| \leq k$, then both sides of the expression are zero. Thus, it suffices to consider the case $|\Lambda| \geq k + 1$. On a discrete set $\Lambda$ with at least $k + 1$ distinct points, $f(x)$ has a best (on $\Lambda$) polynomial approximation $p^*$ of degree $k - 1$ that equioscillates $f(x)$ at $k + 1$ points [60, Theorem 1.11]. That is, there exist values $\lambda_1 < \cdots < \lambda_{k+1}$ contained in $\Lambda$ such that

$$f(\lambda_i) - p^*(\lambda_i) = (-1)^{i-1}\epsilon, \qquad i = 1, \ldots, k + 1,$$

where

$$\epsilon := \max_{x \in \Lambda} |f(x) - p^*(x)|.$$

Similarly, for Item 2 if $f(x)$ is continuous on $\mathcal{I} = [\lambda_{\min}, \lambda_{\max}]$, then $f(x)$ has a unique best polynomial approximation $p^*$ of degree $k - 1$ which equioscillates at $k + 1$ points $\lambda_1 < \cdots < \lambda_{k+1}$ in $\mathcal{I}$ (see for instance [60, 64]), and we analogously define $\epsilon$ as

$$\epsilon := \max_{x \in \mathcal{I}} |f(x) - p^*(x)|.$$

Set

$$\mathbf{A} = \operatorname{diag}(\lambda_1, \ldots, \lambda_{k+1}, \underbrace{\lambda_{k+1}, \ldots, \lambda_{k+1}}_{d-k-1 \text{ times}}), \quad \mathbf{b} = [b_1, \ldots, b_{k+1}, \underbrace{0, \ldots, 0}_{d-k-1 \text{ times}}]^\mathsf{T}.$$

Then,

$$\min_{\deg(p)<k} \|f(\mathbf{A})\mathbf{b} - p(\mathbf{A})\mathbf{b}\|_2^2 = \min_{\alpha_0, \ldots, \alpha_{k-1}} \sum_{i=1}^{k+1} b_i^2 \big(f(\lambda_i) - p(\lambda_i)\big)^2, \qquad p(\lambda) := \sum_{j=0}^{k-1} \alpha_j \lambda^j.$$

We would like to find $\mathbf{b}$ so that $f(\lambda_i) - p(\lambda_i) = (-1)^{i-1}\epsilon$; i.e. so that $p = p^*$. When $\mathbf{b}$ is fixed, for each $j = 0, 1, \ldots, k - 1$ the solution to this least squares problem must satisfy

$$0 = \frac{\mathrm{d}}{\mathrm{d}\alpha_j} \sum_{i=1}^{k+1} b_i^2 \big(f(\lambda_i) - p(\lambda_i)\big)^2 = -2 \sum_{i=1}^{k+1} b_i^2 \left(f(\lambda_i) - p(\lambda_i)\right) \lambda_i^j, \qquad j = 0, 1, \ldots, k - 1.$$

When $p = p^*$, this gives the conditions

$$0 = -2 \sum_{i=1}^{k+1} b_i^2 (-1)^{i-1} \epsilon \lambda_i^j, \qquad j = 0, 1, \ldots, k - 1.$$

Without loss of generality, we can assume $\|\mathbf{b}\|_2 = 1$ so that $b_1^2 + \cdots + b_{k+1}^2 = 1$. Thus, we obtain a linear system

$$\begin{bmatrix} 1 & 1 & \cdots & 1 \\ -2\epsilon\lambda_1^0 & 2\epsilon\lambda_2^0 & \cdots & 2(-1)^{k+1}\lambda_{k+1}^0\epsilon \\ \vdots & \vdots & & \vdots \\ -2\epsilon\lambda_1^{k-1} & 2\epsilon\lambda_2^{k-1} & \cdots & 2(-1)^{k+1}\epsilon\lambda_{k+1}^{k-1} \end{bmatrix} \begin{bmatrix} b_1^2 \\ b_2^2 \\ \vdots \\ b_{k+1}^2 \end{bmatrix} = \begin{bmatrix} 1 \\ 0 \\ \vdots \\ 0 \end{bmatrix}.$$

We can rewrite this system as

$$
\begin{bmatrix}
1 & -1 & \cdots & (-1)^k \\
\lambda_1^0 & \lambda_2^0 & \cdots & \lambda_{k+1}^0 \\
\vdots & \vdots & & \vdots \\
\lambda_1^{k-1} & \lambda_2^{k-1} & \cdots & \lambda_{k+1}^{k-1}
\end{bmatrix}
\begin{bmatrix}
b_1^2 \\
-b_2^2 \\
\vdots \\
(-1)^k b_{k+1}^2
\end{bmatrix}
=
\begin{bmatrix}
1 \\
0 \\
\vdots \\
0
\end{bmatrix}.
$$

This system can be solved analytically via Cramer's rule and has solution

$$
b_\ell^2 = \left( \prod_{\substack{i=1 \\ i\neq\ell}}^{k+1} \prod_{\substack{j=i+1 \\ j\neq\ell}}^{k+1} (\lambda_j - \lambda_i) \right) \Bigg/ \left( \sum_{m=1}^{k+1} \prod_{\substack{i=1 \\ i\neq m}}^{k+1} \prod_{\substack{j=i+1 \\ j\neq m}}^{k+1} (\lambda_j - \lambda_i) \right).
$$

Clearly $(\lambda_j - \lambda_i)$ is positive for $j \in \{i+1, \ldots k+1\}$. Thus the numerator and denominator above are positive, so the entries of $\mathbf{b}$ are well-defined and real. This implies that, for these choices of $\mathbf{A}$ and $\mathbf{b}$,

$$
\min_{\deg(p)<k} \|f(\mathbf{A})\mathbf{b} - p(\mathbf{A})\mathbf{b}\|_2^2 = \min_{\alpha_0,\ldots,\alpha_{k-1}} \sum_{i=1}^{k+1} b_i^2 \big(f(\lambda_i) - p(\lambda_i)\big)^2 = \sum_{i=1}^{k+1} b_i^2 \epsilon^2 = \epsilon^2.
$$

Taking the square root of both sides gives the result. $\qquad\square$

## C.1 Relation between spectrum and instance optimality for random vectors

We also note that, with some additional assumptions on $\mathbf{b}$, spectrum optimality implies instance optimality:

**Lemma 15.** *Let $\{\mathbf{u}_1, \ldots, \mathbf{u}_d\}$ be the eigenvectors of $\mathbf{A}$. Given a problem instance $(f, \mathbf{A}, \mathbf{b}, k)$, if an algorithm is nearly spectrum optimal with parameters $C$ and $c$, then it is nearly instance optimal with parameters $C \cdot \frac{\|\mathbf{b}\|_2}{\min_j |\mathbf{u}_j^\mathsf{T}\mathbf{b}|}$ and $c$.*

The bound is useful e.g., when $\mathbf{b}$ is a random vector. For example, when $\mathbf{b}$ has independent and identically distributed Gaussian entries, $\frac{\|\mathbf{b}\|_2}{\min_j |\mathbf{u}_j^\mathsf{T}\mathbf{b}|} = O(d^{3/2})$ with high probability. Such random vectors occur when sampling Gaussians [59] and are extremely common in machine learning applications related to trace and spectrum approximation [58, 32, 69, 7, 13, 14].

Let $\mathbf{V}$ be the matrix whose $i$th column is $\mathbf{v}_i$. Let $w_i = \mathbf{v}_i^\mathsf{T}\mathbf{b}$ and $\mathbf{w} = \mathbf{V}^\mathsf{T}\mathbf{b}$ be the vector whose $i$th entry is $w_i$. Then,

$$
\|f(\mathbf{A})\mathbf{b} - p(\mathbf{A})\mathbf{b}\|_2^2 = \sum_{i=1}^d (f(\lambda_i) - p(\lambda_i))^2 w_i^2 \geq \left( \min_j w_j^2 \right) \sum_{i=1}^d (f(\lambda_i) - p(\lambda_i))^2.
$$

Clearly $\sum_{i=1}^d (f(\lambda_i) - p(\lambda_i))^2 \geq \max_{x\in\Lambda} |f(x) - p(x)|^2$. Taking the square root of both sides and substituting into Definition 12 proves the result.

For completeness, we now show that if $\mathbf{b} \sim \mathcal{N}(\mathbf{0}, \mathbf{I}_d)$, then with high probability,

$$
\frac{\|\mathbf{b}\|_2}{\min_j |\mathbf{v}_j^\mathsf{T}\mathbf{b}|} = O(d^{3/2}).
$$

First, with high probability, $\|\mathbf{b}\|_2 = \Theta(\sqrt{d})$; see, for example, [66, p. 3.1.2]. Second, we show that with high probability $|\mathbf{v}_j^\mathsf{T}\mathbf{b}| = \Omega(1/d)$ for all $j$. Since $\mathbf{V}$ is orthonormal, $\mathbf{w} := \mathbf{V}^\mathsf{T}\mathbf{b} \sim \mathcal{N}(\mathbf{0}, \mathbf{I}_d)$. By anti-concentration of the normal distribution, the probability that $|w_j| > \epsilon$ is at least $1 - 0.4\epsilon$ for any $\epsilon$. By a union bound, this holds simultaneously for all $j$ with probability at least $1 - 0.4\epsilon d$. Setting $\epsilon = \Theta(1/d)$ finishes the argument. Finally, by another union bound, our bounds on the numerator and on the denominator hold simultaneously with high probability.

# D    Proofs of Theorem 6 and Theorem 7

We begin by quoting bounds from [10]:

**Lemma 16.** *For all $k \geq 1$, the Lanczos-FA iterate satisfies the bounds*

*1.* $\|\mathbf{A}^{1/2}\mathbf{b} - \mathsf{lan}_k(x^{1/2}; \mathbf{A}, \mathbf{b})\|_2 \leq \dfrac{\lambda_{\max}^{3/2}}{2k^{3/2}} \cdot \|\mathbf{A}^{-1}\mathbf{b} - \mathsf{lan}_k(1/x; \mathbf{A}, \mathbf{b})\|_2.$

*2.* $\|\mathbf{A}^{-1/2}\mathbf{b} - \mathsf{lan}_k(x^{-1/2}; \mathbf{A}, \mathbf{b})\|_2 \leq \sqrt{\dfrac{\lambda_{\max}}{\pi k}} \cdot \|\mathbf{A}^{-1}\mathbf{b} - \mathsf{lan}_k(1/x; \mathbf{A}, \mathbf{b})\|_2.$

*Proof.* For Part 1, see Example 4.1 in [10]. For part Part 2, we use the same proof again but substituting the inverse square root for the square root. For both parts, we have simplified the resulting bounds using the fact that for $k \geq 1$,

$$\frac{\Gamma(k - 1/2)}{\Gamma(k + 1)} \leq \frac{\sqrt{\pi}}{k^{3/2}} \qquad\qquad \frac{\Gamma(k + 1/2)}{\Gamma(k + 1)} \leq \frac{1}{\sqrt{k}}.$$

□

Next, we bound the error of Lanczos-FA on linear systems by the error of the optimal polynomial approximation to $x^{-1/2}$:

**Lemma 17.** *For all $k \geq 1$, the Lanczos-FA iterate satisfies the bounds*

$$\|\mathbf{A}^{-1}\mathbf{b} - \mathsf{lan}_k(1/x; \mathbf{A}, \mathbf{b})\|_2 \leq \frac{3}{\sqrt{\lambda_{\min}}} \sqrt{\kappa(\mathbf{A})} \|\mathbf{b}\|_2 \min_{\deg(p) < k/2} \left( \max_{x \in \Lambda} \left| \frac{1}{\sqrt{x}} - p(x) \right| \right).$$

*Proof.* Let

$$p^*(x) = \operatorname*{argmin}_{\deg(p) < k/2} \max_{x \in \Lambda} \left| \frac{1}{\sqrt{x}} - p(x) \right|$$

and note that $p(x)^2$ is a polynomial of degree less than $k$. Therefore,

$$\min_{\deg(p) < k} \|\mathbf{A}^{-1}\mathbf{b} - p(\mathbf{A})\mathbf{b}\|_2 \leq \|\mathbf{b}\|_2 \min_{\deg p < k} \max_{x \in \Lambda} \left| \frac{1}{x} - p(x) \right| \leq \|\mathbf{b}\|_2 \max_{x \in \Lambda} \left| \frac{1}{x} - p^*(x)^2 \right|.$$

Thus, since $|1/x - p^*(x)^2| = |1/\sqrt{x} + p^*(x)| \cdot |1/\sqrt{x} - p^*(x)|$, we can plug the above into (17) to obtain a bound

$$\|\mathbf{A}^{-1}\mathbf{b} - \mathsf{lan}_k(1/x)\|_2 \leq \sqrt{\kappa(\mathbf{A})} \|\mathbf{b}\|_2 \left( \max_{x \in \Lambda} \left| \frac{1}{\sqrt{x}} + p^*(x) \right| \right) \left( \max_{x \in \Lambda} \left| \frac{1}{\sqrt{x}} - p^*(x) \right| \right).$$

Now use can use the optimality of $p^*(x)$ in approximating $1/\sqrt{x}$ on $\Lambda$ to bound:

$$\max_{x \in \Lambda} \left| \frac{1}{\sqrt{x}} + p^*(x) \right| = \max_{x \in \Lambda} \left| \frac{2}{\sqrt{x}} + p^*(x) - \frac{1}{\sqrt{x}} \right|$$

$$\leq \max_{x \in \Lambda} \left| \frac{2}{\sqrt{x}} \right| + \max_{x \in \Lambda} \left| p^*(x) - \frac{1}{\sqrt{x}} \right|$$

$$\leq \frac{2}{\sqrt{\lambda_{\min}}} + \max_{x \in \Lambda} \left| 0(x) - \frac{1}{\sqrt{x}} \right| \leq \frac{3}{\sqrt{\lambda_{\min}}}.$$

Substituting this inequality above proves the lemma.                                                    □

The main proofs now follow directly.

*Proof of Theorem 6.* Substitute Lemma 17 into Lemma 16, part (b).                          □

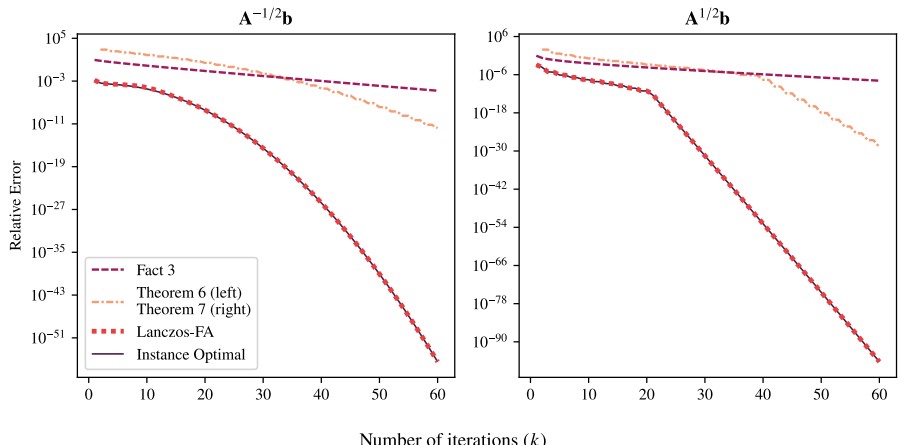

**Figure 7:** The bounds of Theorems 6 and 7 capture the convergence behavior of Lanczos-FA for $\mathbf{A}^{\pm 1/2}\mathbf{b}$. In particular, they can be tighter than the FOV optimality bound of Fact 3. The predicted rate of convergence is about half that observed for Lanczos-FA in this example; this is due to the parameter $c = 1/2$ in both bounds.

*Proof of Theorem 7.* Let

$$p^*(x) = \underset{\deg(p) < k/2+1}{\operatorname{argmin}} \max_{x \in \Lambda \cup \{0\}} \left| \sqrt{x} - p(x) \right|, \qquad \tilde{p}(x) = \frac{p^*(x) - p^*(0)}{x}$$

and note that $\tilde{p}$ is a polynomial of degree less than $k/2$. Then

$$
\begin{aligned}
\min_{\deg(p) < k/2} \left( \max_{x \in \Lambda} \left| \frac{1}{\sqrt{x}} - p(x) \right| \right) &\leq \max_{x \in \Lambda} \left| \frac{1}{\sqrt{x}} - \tilde{p}(x) \right| \\
&= \max_{x \in \Lambda} \left( |1/x| \cdot \left| \sqrt{x} - x\tilde{p}(x) \right| \right) \\
&\leq \frac{1}{\lambda_{\min}} \max_{x \in \Lambda} \left| \sqrt{x} - p^*(x) + p^*(0) \right| \\
&\leq \frac{1}{\lambda_{\min}} \max_{x \in \Lambda} \left( \left| \sqrt{x} - p^*(x) \right| + \left| \sqrt{0} - p^*(0) \right| \right) \\
&\leq \frac{2}{\lambda_{\min}} \max_{x \in \Lambda \cup \{0\}} \left| \sqrt{x} - p^*(x) \right|.
\end{aligned}
$$

Combining Lemma 16 part (a), Lemma 17, and the above proves the theorem. □

## E  Additional Experiments

### E.1  Validating Theorems 6 and 7

To understand Theorems 6 and 7, we repeat the experiment using the inverse square root and square root functions. We reuse the second and third $\mathbf{A}$ matrix and $\mathbf{b}$ vectors from the previous experiment. As Figure 7 shows, our bounds are tighter than Fact 3. They closely resemble the true convergence of Lanczos-FA, but stretched horizontally by a factor of 2. This is because the degree of the polynomial minimization in both bounds is $k/2$, while in these examples Lanczos-FA performs nearly instance optimally (that is, as well as the best degree $k$ polynomial). Still, our bounds capture the correct qualitative behavior of the algorithm.

### E.2  Rational functions with poles between eigenvalues

In Theorem 4, we assume that the poles of the rational function $r(x)$ lie outside the interval $\mathcal{I} = [\lambda_{\min}, \lambda_{\max}]$ containing $\mathbf{A}$'s eigenvalues. If the poles of $r(x)$ live in $\mathcal{I}$, then the Lanczos-FA iterate

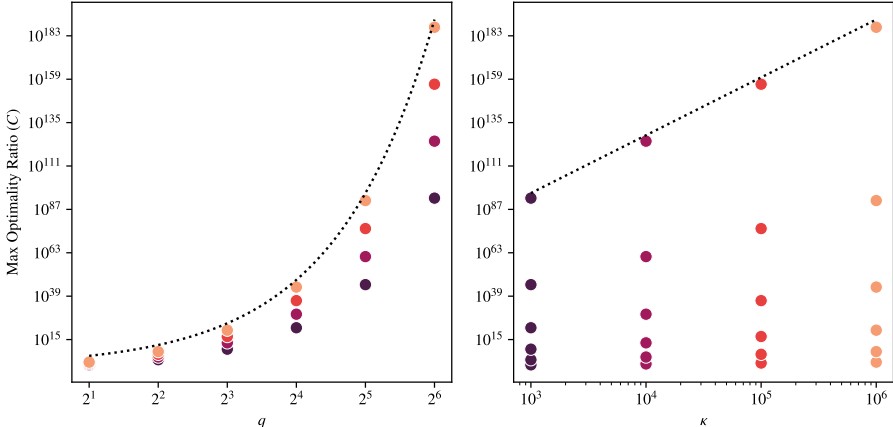

**Figure 8:** The maximum observed ratio between the error of Lanczos-OR and the optimal error over choice of right hand side **b** when approximating $\mathbf{A}^{-q}$ for matrices with varying condition number $\kappa$. Each point shows the optimality ratio for a different pair of $\kappa$ and $q$. Points with the same color correspond to the same value of $\kappa$. On the left, the dotted line plots $g(q) = \kappa^{q/2}$ for the maximum $\kappa$ considered ($10^6$). On the right, the dotted line plots $g(\kappa) = \kappa^{q/2}$ for the maximum $q$ considered ($2^6 = 64$). Overall, the optimality ratio appears to grow as $\Omega(\kappa^{q/2})$. Contrast with Figure 3.

can be arbitrarily far from the optimal iterate. Indeed, an eigenvalue of **T** might be very close to a pole, causing $f(\mathbf{T})$ to be poorly behaved. This behavior is easily observed e.g. for $f(\mathbf{A}) = \mathbf{A}^{-1}$ and rules out a direct analog of Theorem 4 when $r(x)$ has poles in $\mathcal{I}$. However, as noted in (18), for $1/x$ there exists bounds for Lanczos-FA in terms of the best possible KSM [12].

However, as discussed in Appendix B.1, the "overall" convergence of Lanczos-FA still seems to closely follow the optimal approximation for many matrix functions. Specifically, even if the Lanczos-FA iterate deviates significantly from optimal on some iterations, there are other iterations where it is very close to optimal. We illustrate this phenomena in Figure 6 for two rational functions with poles in $\mathcal{I} = [\lambda_{\min}, \lambda_{\max}]$, as well as the function $\text{sign}(x)$, which has a discontinuity in $\mathcal{I}$ and would typically be approximated by a rational function with conjugate pairs of imaginary poles on the imaginary axis which become increasingly close to the the interval as the rational function degree increases.[8]

### E.3 Comparison with Lanczos-OR

As discussed in Appendix B.2, the Lanczos-OR method is an alternative method for rational matrix function approximation for which a tighter guarantee than Theorem 4 is currently known (the iterates are optimal in a certain norm). However, Lanczos-FA usually outperforms Lanczos-OR when measured in the 2-norm. For example, as discussed in Section 4.1, when approximating $\mathbf{A}^{-q}\mathbf{b}$, the worst optimality ratio of Lanczos-FA that we were able to observe was $O(\sqrt{q \cdot \kappa(\mathbf{A})})$, much lower than Theorem 4 would suggest). Repeating the same experiment with Lanczos-OR shows that the method's optimality ratio appears to grow as $\Omega(\kappa(\mathbf{A})^{q/2})$ (see Figure 8).

---

[8]We remark that [11] shows that applying their Lanczos-OR method to each term in the partial fraction decomposition of the approximating rational function yields an approximation to the sign function that seems to avoid the oscillations of Lanczos-FA.

