# OpenReview forum: "Nearly Optimal Approximation of Matrix Functions by the Lanczos Method"
_NeurIPS.cc/2024/Conference — NeurIPS 2024 spotlight_

### Official Review · Reviewer_dxxb · 2024-06-30

**Soundness:** 3
**Presentation:** 3
**Contribution:** 2
**Rating:** 6
**Confidence:** 2

**Summary:**

The authors investigate the convergence of the Lanczos method for computing $f(A)v$ for a matrix $A$ and a vector $v$. They show a near instance optimality bound that involves the product of the condition numbers of matrices related to $A$ and empirically investigate the bound.

**Strengths:**

The authors investigate the convergence of the Lanczos method for computing matrix functions theoretically and numerically. The results are also meaningful for the machine learning community since matrix functions appear in many machine learning algorithms.

**Weaknesses:**

I think the authors should explain more about how we can interprete the proposed bound. The authors insist the bound captures the convergence behavior of the Lanczos method. The proposed bound involves the condition numbers of the matrices appearing in the denominator of the rational function. I think it is natural, but I couldn't clearly understand why the proposed bound can explain the convergence behavior better than the existing bound in Fact 1,  since if the condition numbers of these matrices are large, than the magnitude of $f$ in the bound in Fact 1 can be large, and the factor involving $f$ can also become large.

**Questions:**

- Can you generalize your results to other types of Krylov subspace methods such as rational Krylov methods?
- In the middle and the right figures in Figure 1, the relative error seems very small. Did you use higher precision floating point format than double precision?

**Limitations:**

The authors discuss limitations in Section 4.

---

> ### Author Rebuttal · Authors · 2024-08-06
>
> Thank you for your thoughtful feedback. We will respond to each of the questions in turn:
>
> > “Can you generalize your results to other types of Krylov subspace methods such as rational Krylov methods?”
>
> It would be interesting to see whether bounds of the flavor we describe can be extended to rational Krylov subspace methods. However, the class of functions we are studying are already rational functions. If one were to use a rational Krylov subspace method (with a suitably chosen set of poles), these functions could be applied exactly. So it is somewhat less clear exactly what the generalization would look like.
>
> > “In the middle and the right figures in Figure 1, the relative error seems very small. Did you use higher precision floating point format than double precision?”
>
> Yes, we used higher precision arithmetic. The precise setup is described in the first paragraph of Section 3.
>
> > “I couldn't clearly understand why the proposed bound can explain the convergence behavior better than the existing bound in Fact 1”
>
> It is true that our main bound (Theorem 1) is weaker if the condition number of each $A_i$ is large. This means that, as Figure 2 shows, it is sometimes better than Fact 1 and sometimes worse. See the discussion below Theorem 1, at the bottom of page 4. However, importantly, while it depends on the condition numbers, our leading constant factor $C$ does *not* depend on the number of Lanczos iterations. On the other hand, the term $\min_{\mathrm{deg}(p) < k-q+1} \||r(A)b - p(A)b\||_2$ in Theorem 1 does: it decreases with the number of iterations, $k$, typically at a much faster rate than Fact 1. For this reason, Theorem 1 usually beats Fact 1 eventually as the number of iterations increases. If our current $C$ factor can be improved (we suspect it can be), then Theorem 1 would beat Fact 1 earlier.
>
> As an extreme example, it might help to consider the simplified case of a 2x2 matrix with only two eigenvalues: $\lambda_{\min}$ and $\lambda_{\max}$. Let’s compare Definition 2, which is the form of our main theorem, to Fact 1. Both of them upper bound the error by a minimization problem over polynomials of a certain degree. To get a good guarantee from Fact 1, we would have to find a degree-$k$ polynomial that closely matches $f$ on the entire range $[\lambda_{\min}, \lambda_{\max}]$. When the condition number of $A$ is large, this range will be large, and so this will be impossible. However, to get a good guarantee from Definition 2, we only need to find a polynomial that matches $f$ at two discrete points: $\lambda_{\min}$, and $\lambda_{\max}$. This is much easier to do. In fact, if $k \geq 1$, then our bound guarantees that we get zero error for any $\lambda_{\min}$, $\lambda_{\max}$, and any $f$. Fact 1 does not capture this behavior.

---

> > ### Comment · Reviewer_dxxb · 2024-08-11
> >
> > Thank you for the rebuttal. After reading the rebuttal and reviews from other reviewers, I decided to raise my score.

---

### Official Review · Reviewer_uopa · 2024-07-08

**Soundness:** 3
**Presentation:** 3
**Contribution:** 3
**Rating:** 7
**Confidence:** 2

**Summary:**

A matrix function f(A) for a real, symmetric matrix A and a univariate function f is given as \sum f(\lambda_i) u_i u_i^\top, where \lambda_i are the eigenvalues of A and u_i the corresponding eigenvectors. In practice, one is often interested in matrix-vector products f(A)b for some vector b. In the paper, the approximation quality of the Lanczos-FA method for approximating f(A)b for rational functions f is analyzed theoretically and experimentally. In particular, a known bound that is uniform in the A and b is improved to instance specific bounds using the notion of near instance optimality. The instance specific bounds are not tight. However, they can better explain better the experimentally observed good approximation quality of the Lanczos-FA method that often outperforms more advanced Krylov subspace methods.

**Strengths:**

Originality: The paper provides new theoretical insights into the observed practical behavior of the Lanczos-FA method. It contributes to a better understanding of Krylov subspace methods. The results are non-trivial.

Clarity:  The paper is well is structured and, in general, easy to read. I did not read the full proof of Theorem 1, but appreciate the proof sketch in Section 2.1.

Significance: As far as I can tell, the paper addresses an interesting, practically relevant problem. The paper does not only present upper bounds but also aims to find instances for the Lanczos-FA.

**Weaknesses:**

As far as I can tell, there are no major weaknesses.

Minor typos and suggestions for improvement:

Introduction: Bayseian

Section 1.4: ... functions of interest(ing) ...

Footnote 2: Should \|x\|_A be \|b\|_a?

Line 148: The (second factor) of the second term ...

Figure 2: The axis labels are hardly legible.

Figure 5: matvec -> matrix-vector product

**Questions:**

1. I like the proof sketch in Section 2.1, which was easy to follow for a non-expert like me. The only thing that I did not see immediately was the equality below Line 145. It is probably an elementary result, but maybe you can provide a few more insights why this equality holds true.

Is well addressed in the author's rebuttal.

2. In the second paragraph of Section 3 (Experiments): Why did you choose the matrices (or the spectra of the matrices) as you did? How do practical spectra look like? What are extreme/challenging spectra?

Is also well addressed in the author's rebuttal.

**Limitations:**

As far as I can tell, there are no limitations. The bounds are not tight, but I do not consider this a limitation of the paper.

---

> ### Author Rebuttal · Authors · 2024-08-06
>
> Thank you for the thoughtful feedback! We will address the two questions in turn:
>
> The first asks for clarification about Line 145. The projection of a vector $x$ onto a linear subspace $S$ is defined as $\underset{y \in S}{\operatorname{argmin}} \||y - x\||_2$. The $A$-norm projection is defined as $\underset{y \in S}{\operatorname{argmin}} \||y - x\||_A = \underset{y \in S}{\operatorname{argmin}} \||A^{1/2} y - A^{1/2} x\||_2$. In our case, the linear subspace is the Krylov subspace $K_k(A,b)$. By definition, $Q$ is a basis for this subspace. Therefore, any $y \in K_k(A,b)$ can be written as $Qc$ for some $c$. Thus, the $A$-norm projection can be written $\underset{c}{\operatorname{argmin}} \||A^{1/2} Q c - A^{1/2} x\||_2$. Solving this least squares problem using the normal equations, we find that $c = (Q^\top A Q)^{-1} Q^\top A x = T^{-1} Q^\top A x$. Therefore, the projection is $y = Qc = Q T^{-1} Q^\top A x$. In other words, $\||x - Q T^{-1} Q^\top A x \||_2 = \underset{y \in K_k}{\operatorname{min}} \||x - y\||_2$. Finally, by definition, any vector in the Krylov subspace can be written as $p(A)b$ for some polynomial $p$ with degree $< k$, so we have $\underset{y \in K_k(A, b)}{\operatorname{min}} \||x - y\||_2 = \underset{p}{\operatorname{min}} \||x - p(A)x\||_2$. To finish the argument, replace $x$ with $A^{-2}b$. We will modify the proof sketch to clarify this step.
>
> The second asks about the choice of matrices for our experiments. First, note that all Krylov subspace methods are equivariant to orthogonal transformations, and the errors $\||f(A)b - \mathrm{alg}\||$ are invariant to such transforms (in the Euclidean norm). So, without loss of generality, we choose $A$ to be a diagonal matrix, as is standard in the literature.
>
> It is well understood in the literature that the eigenvalue distribution has a large influence on the performance of Krylov subspace methods, often in subtle ways. For instance, see reference [8], “Towards understanding CG and GMRES through examples”. We used uniform, skewed, and clustered eigenvalue distributions to get some variety in the convergence behavior (as reflected in the different shapes of the curves in Figure 2). In particular, eigenvalue distributions that are clustered at the ends of the spectrum often have interesting behavior. A non-uniform distribution can emphasize the difference between the uniform optimality of Fact 1 and the instance optimality of our bounds. As we report in Section 3.1, distributions with a single outlying eigenvalue are among the most challenging for Lanczos-FA.
>
> What practical spectra look like depends a lot on the specific problem, but they are often characterized by the presence of many small eigenvalues and a smaller number of large outlying and clustered eigenvalues (hence, why we consider skewed and clustered eigenvalue distributions).
>
> We also thank you for spotting these typos. They will all be corrected in the revision.

---

> > ### Comment · Reviewer_uopa · 2024-08-09
> >
> > Thank you for answering my questions. I like your paper. To make a clear statement, I will increase my score to accept.

---

### Official Review · Reviewer_nxLf · 2024-07-11

**Soundness:** 3
**Presentation:** 3
**Contribution:** 3
**Rating:** 6
**Confidence:** 4

**Summary:**

The submission discusses the optimality (in terms of approximation error) of approximation $(A, b) \rightarrow f(A)b$ with Lanczos' algorithm.
The main result (Theorem 1) states that the Lanczos iteration is "near instance optimal". Near-instance-optimality relates to the optimal reconstruction of $r(A)b$ for rational functions $r$.
Via the triangle inequality, one can deduce similar statements for general $f$ provided $f$ can be approximated well by rational functions.
The submission proves Theorem 1 by repeatedly applying a similar optimality result for $A^{-1} b$ (i.e. $f=1/x$), which is why the final bound comes with scalar $\kappa(A)^q$ where $\kappa$ is the condition number of $A$ and $q$ the order of the denominator of the rational function.

**Strengths:**

The paper provides an interesting perspective on the performance of Lanczos' algorithm for computing matrix functions.
The main strength is clarity, partly because the result is relatively strong and partly because the presentation is easy to follow.
I appreciate the proof sketch in Section 2.1 and the comparison to prior work in Appendix B (especially in B.2, which I was wondering about while reading the other parts of the manuscript).
I also appreciate the code submission.
Overall, I think this is a nice paper.

**Weaknesses:**

The paper's main contribution is Theorem 1: near-instance optimality of Lanczos' method for computing $f(A)b$.
The weaknesses of this result are twofold:
1. The analysis is limited to exact arithmetic. It is well-known that implementations of Lanczos' method in exact versus finite-precision arithmetic may not match. Focussing on exact arithmetic somewhat limits the practical applicability of Theorem 1.
2. The bound in Theorem 1 includes the constant $\kappa(A)^q$. Similar works (including those discussed in Appendix B) achieve smaller constants, and in practical applications, $\kappa(A)$ can be gigantic. Since the largest $\kappa$ in the experiments seems to be $10^6$ (unless I have missed something), it is slightly unclear whether the bounds actually do explain the superiority of Lanczos' algorithm in practice (where $\kappa$ can exceed $10^6$ by a large margin; see, for instance,  the condition number of Gaussian process covariance matrices or discretisations of PDEs).

That said, both weaknesses are openly discussed as limitations.

Furthermore, Krylov methods have gained popularity in machine learning in recent years, as evidenced by the growing number of papers/submissions on Lanczos, Arnoldi, CG, GMRES, etc.
Nevertheless, the submission's connection to existing machine learning literature would be strengthened if at least some of the matrix functions mentioned in the introduction would reappear in the experiments.

**Questions:**

What follows is not directly a question, but the submission's format appears to deviate from the Neurips style in two ways:

1. The abstract contains two paragraphs, even though the template states, "The abstract must be limited to one paragraph."
2. It seems that the mathematics font has been changed from the original Neurips template.

I am unsure what Neurips' policy is here. Should these two things be changed?

**Limitations:**

Limitations are acknowledged. (I discuss this under "Weaknesses" above.)

---

> ### Author Rebuttal · Authors · 2024-08-06
>
> Thank you for the thorough and thoughtful comments. This review raises several valuable points:
>
> On the issues of exact vs. floating point arithmetic, please see the global response.
>
> Regarding the leading constant in Theorem 1, it is true that our result is weaker if the condition number of each $A_i$ is large. This means that, as Figure 2 shows, it is sometimes better than existing bounds (Fact 1) and sometimes worse. See the discussion below Theorem 1, at the bottom of page 4. However, as discussed in Section 4, we believe that the dependence on the condition number in the leading constant $C$ can likely be improved significantly with a tighter theoretical analysis, bringing the bound closer to more specialized results (for specific matrix functions), like those discussed in Appendix B.1.
>
> The review notes that it would be valuable for the paper to touch on the matrix functions mentioned in the introduction. While our main focus is rational functions, we do study several of the functions most relevant for machine learning both theoretically and empirically:
> - Matrix square root and inverse square root: Analyzed theoretically in Appendix D. Experiments in Appendix E.1. Also discussed below Lemma 1.
> - Matrix exponential: For theoretical analysis, see the discussion below Lemma 1. Experiments are shown in the right panel of Figure 1. A rational approximation to it is studied in Figure 2. See the discussion accompanying these figures at the beginning of Section 3.
> - Matrix log: Experiments are shown in the right panel of Figure 1. A rational approximation to it is studied in Figure 2. See the discussion accompanying these figures at the beginning of Section 3.
> - Matrix sign: This is discussed at the end of Section 3.2. Results are shown in Figure 5 and Figure 6
>
> Thank you for spotting the deviations from the NeurIPS style. These were oversights and we are more than happy to correct them for the revision.

---

> > ### Comment · Reviewer_nxLf · 2024-08-09
> >
> > Thank you for iterating!
> >
> > I agree with your perspective on exact versus finite-precision arithmetic. Nonetheless, assuming exact arithmetic remains a weakness of the analysis (though typical for this type of work).
> > I also agree with the rebuttal's discussion on the constant in Theorem 1. However, this constant also remains a weakness of the theory, even though there is hope that the bound can be tightened.
> >
> > Finally, about the matrix function experiments: my review might have been poorly phrased; apologies for that. What it referred to was not that the matrix function was missing but its application. For example, the current experiments include log-determinant estimation in Figure 1, but only on a toy matrix, whereas a case study in Gaussian process regression would strengthen the paper's connection to the ML literature. (Same for the matrix sign or  matrix exponential, for example.)
> >
> > Altogether, I continue to recommend accepting the submission. Thank you again for the reply.

---

### Official Review · Reviewer_CAYW · 2024-07-12

**Soundness:** 4
**Presentation:** 4
**Contribution:** 4
**Rating:** 8
**Confidence:** 3

**Summary:**

The authors study computation of matvecs of a matrix function via the Lanczos method, an in particular try to answer the question of why the basic Lanczos method is competitive or even superior to sophisticated implementations targeting specific matrix functions. They provide theoretical bounds on the error of Lanczos methods for rational functions which they use to produce a bound on generic functions based on distance to the nearest rational, and demonstrate this in numerical experiments.

**Strengths:**

The first major contribution of this article lies in the introduction section, which is establishment of the fact that Lanczos methods outperform sophisticated, problem specific peers in some instances. This "theory-practice gap" in and of itself is an important and consequential fact to establish. The numerical experiments they do to demonstrate this are convincing and well communicated.

Subsequently, they offer important theorems governing the ability of Lanczos methods to approximate rational functions, and use this to find a bound based on the l infinity distance between a given function and the nearest rational function of low degree. Numerical experiments demonstrate these results on important practical cases.

The way that the main theorem was sketched as special case in Section 2.1 is an exceptional move from a mathematical communication perspective. I was easily able to follow that specific proof, and it gave me intuition as to why the general case might be true without going through the more extensive proof in the appendix. I wish more articles did this, and I will try to do this in my future articles.

**Weaknesses:**

In practice, we care about the effectiveness of this method in floating point arithmetic. But I think it makes sense to partition that off as a separate project, as the contents of the article as it stands are convincing on their own.

Here are some grammar issues I noticed (did not affect my scoring):
33) functions repeated.
72) an problem instance -> a problem instance
146) the "and <=" may be missing a term.
147 and 148) May be worth clarifying that it is the right hand side of (6) that is being referred to.

**Questions:**

Congratulations on this paper.

**Limitations:**

The theoretical results rely on the poles of the rational function lying outside of the spectrum of the operator which might not always be the case, but for many practical situations it will be, and the authors' assertion that this case be considered future work is convincing to me.

---

> ### Author Rebuttal · Authors · 2024-08-06
>
> Thank you for the thorough review! The point about floating point arithmetic is well-taken; see the global response for more. Thank you as well for the grammar corrections. These have all been corrected in our revision.

---

### Official Review · Reviewer_cS79 · 2024-07-12

**Soundness:** 4
**Presentation:** 4
**Contribution:** 3
**Rating:** 7
**Confidence:** 3

**Summary:**

The goal of this paper is to better understand the performance of using Lanczos for matrix-function times vector computations. Using Lanczos for this task is the de-facto standard, since it works very well (best vs competitors) in practice. Previous theory is very weak: there is a big gap between the bounds it specifies and the actual performance. The goal of this paper is shrink and where possible eliminate this gap.

A stronger bound for Lanczos is established.  Extensive experiments are reported.

**Strengths:**

Very well written paper.

The main value of the paper is that it takes a well established method and de-facto standard (using Lanczos for matrix-function times vector computations), but with weak theory, and establishes much stronger theory. The reason this is important is that the weak bound motivated researchers to find better algorithms. Such algorithms had better theoretical bounds, but in practice were not as good as Lanczos. This paper closes the gap, giving stronger theory for Lanczos.

The experimental evaluation is robust. Appendix also contains additional bounds specific for matrix square root and inverse square root.

**Weaknesses:**

- Bounds are theoretic, and only explain what an existing algorithm achieves. There is no new algorithm in the paper.
- The assumption that the poles are outside the gap that contains the eigenvalues seem to imply that the bound works only for a very restricted subset of rational functions. How common is it?
- Bounds are only for exact arithmetic. The picture for finite-precision arithmetic can be very different. Even though I can understand why bound for finite-precision arithmetic are outside the scope for this paper, the authors could have conducted experiments with them to see how much the bounds carry to finite-precision.
- Even though the problem of computing f(A)b has many ML applications, at the core the paper is focused on the quality of a NLA problem (solving f(A)b). It is more a NLA paper than a ML paper.
- In some experiments, in order to show significant gap between the new bound and the old bound (Fact 1), a large number of iteration are used so the error becomes very very small. For example, in Figure 2 (center) the error starts at 10^-5 and goes to 10^-69. In Figure 2 (right) errors start at 10^-10!  In applications you will rarely work in these regimes, and will do with much smaller error. It is unclear whether the gap between Fact 1 and the new bound is significant for that regime.
- Lemma 1 seems a bit weak: the multiplicative factor or norm(b) is C_r which can be huge. Sure, it is present in (7), but there the minimum it multiplies presumed to be very small. In Lemma 1 it appears on norm(b), which is constant.

**Questions:**

- How hard is it to build rational approximation with poles outside the interval of eigenvalues? Can you give a reference to a general result?
- Did you conduct experiments with finite precision to see how much the theory carries over?

**Limitations:**

Nothing to add.

---

> ### Author Rebuttal · Authors · 2024-08-06
>
> Thank you for the thorough and thoughtful comments! This review raises many valuable points:
>
> Regarding the difference between exact arithmetic and finite precision, please see the global response.
>
> The review asks if it is realistic to build a rational approximation with poles outside the interval of eigenvalues. For many functions (e.g. exponential, $x^{\alpha}$) explicit rational approximations with poles outside of the interval of eigenvalues are known. (See for instance, reference 36 and 62.) For other “nice” (e.g. continuous) functions, one could apply the Remez algorithm to find the best rational approximation with real poles (see reference 63, chapter 24-5). The resulting approximation will not have any pole inside the interval of approximation, because otherwise the approximation would have infinite error near that pole. For less nice functions (e.g. absolute value which has a discontinuous derivative), the best rational approximations actually have *complex* poles which are clustered about discontinuities in the (higher order) derivatives. For instance, for the absolute value, the best rational approximation has poles on the imaginary axis. Our theory does not apply to this case, but this would be one of the most natural classes of functions to explore next.
>
> For experiments, we chose to run Lanczos in extended precision because it allows us to highlight the qualitative behavior of our bound: Theorem 1 captures the correct shape of the convergence curve better than Fact 1. Nevertheless, several of our plots (e.g., the left and center panels of Figure 2) show that our analysis can still significantly outperform Fact 1, even when the desired accuracy or number of iterations is low. For some problems, this is not the case, mainly because of the large prefactor in Theorem 1, which depends on the condition number of $A_1, \ldots, A_q$. Such a prefactor does not appear in Fact 1, so our new bound is only better for a large number of iterations. That said, as discussed in Section 4, we believe the prefactor can likely be reduced with a better theoretical analysis, in which case the improvement of Theorem 1 would kick in sooner.
>
> The review also asks about the multiplicative factor in Lemma 1. It is correct to note that in Equation 8, $\||b\||_2$ is a constant. However, in this term, $C_r$ is *also* multiplied by $\||f - r\||_I$, which denotes the approximation error in the max-norm for the rational function approximation $r$. As the degree of $r$ grows, this goes to zero, often much more quickly than the error of the best polynomial approximation to $f$ on the interval $I$.

---

> > ### Comment · Reviewer_cS79 · 2024-08-13
> >
> > Thank you for the answers. I think some form of them should be included in the revised manuscript.
> >
> > My score was already positive before (7 - accept), and stays the same after reading the rebuttal.

---

### Author Rebuttal · Authors · 2024-08-06

We thank the reviewers for their thoughtful comments. We were pleased with the quality and number of reviews we received, and have taken the feedback into account in order to improve the paper. In our global response we address some common points raised by the reviewers.

Several of the reviewers raised very reasonable questions about the applicability of our bounds to finite precision arithmetic. Generally speaking, the behavior of Lanczos in finite precision arithmetic is significantly more complicated than exact arithmetic. Therefore, bounds for exact arithmetic, such as the ones in this paper, serve as a starting point for a more complete understanding of the algorithm in finite precision arithmetic. We do observe that, typically, Lanczos-FA continues to perform near-optimally, even when run with the standard 16 digits of precision. Concretely, plots like those in Figure 1 and Figure 2 can be reproduced in finite precision. We will consider adding such plots to our revised version of the paper (as suggested by Reviewer cS79) to highlight this point.

We make three additional comments:

- First, we note that reorthogonalization is often used in practice. In this case, the behavior of Lanczos in finite precision is very similar to exact arithmetic and we expect our bounds would carry over with a little work.
- Second, even without reorthogonalization, there is some hope our theory can be applied. In particular, the analysis of Greenbaum (ref [32]) guarantees that the behavior of Lanczos in finite precision arithmetic is equivalent (in a precise way) to the behavior of Lanczos in exact arithmetic on a certain related problem. This allows exact arithmetic theory (e.g., our bounds) to be transferred to finite precision arithmetic. We discuss this in the final paragraph of Section 4. However, the precise equivalence is somewhat complicated. This would be an interesting direction for further work.
- Finally, we note that the analogous optimality guarantees for conjugate gradient in exact arithmetic are generally regarded as very important (even if they do not hold precisely in finite precision), because they provide intuition about how the algorithm works and are predictive of its excellent performance in practice.

---

### Decision · Program_Chairs · 2024-09-25

**Decision:**

Accept (spotlight)

**Comment:**

The presented paper studies the approximation of the matrix function using a Lanczos-based method. It shows that, for a natural class of rational functions, Lanczos matches the error of the best possible Krylov subspace method, up to a multiplicative approximation factor.

Five reviewers evaluated the paper, and their overall assessment is positive. I agree with their evaluation and believe the paper makes a significant contribution with compelling results. In particular, I think the presentation of the paper is truly exceptional. Some reviewers raised concerns about the applicability of the proposed bounds in finite precision arithmetic, which the authors addressed satisfactorily.